# Snow depth mapping from stereo satellite imagery in mountainous terrain : evaluation using airborne laser scanning data

César Deschamps-Berger[1,2], Simon Gascoin[1], Etienne Berthier[3], Jeffrey Deems[4], Ethan Gutmann[5], Amaury Dehecq[6,7], David Shean[8], Marie Dumont[2]

[1] Centre d'Etudes Spatiales de la Biosphère, CESBIO, Univ. Toulouse, CNES/CNRS/INRA/IRD/UPS, 31401 Toulouse, France

[2] Université Grenoble Alpes, Université de Toulouse, Météo-France, Grenoble, France, CNRS, CNRM, Centre d'Etudes de la Neige, Grenoble, France

[3] Centre National de la Recherche Scientifique (CNRS-LEGOS), 31400 Toulouse, France

[4] National Snow and Ice Data Center, Boulder, CO, USA

[5] Research Applications Lab, National Center for Atmospheric Research (NCAR), Boulder, CO, USA

[6] Laboratory of Hydraulics, Hydrology and Glaciology (VAW), ETH Zurich, Zurich, Switzerland

[7] Swiss Federal Institute for Forest, Snow and Landscape Research (WSL), Birmensdorf, Switzerland

[8] University of Washington, Dept. of Civil and Environmental Engineering, Seattle, WA

Keywords: seasonal snow, remote sensing, snow depth, snow hydrology, Pléiades, DEM, photogrammetry

*Correspondence to:* _cesar.deschamps-berger@cesbio.cnes.fr_

**Abstract.** An accurate knowledge of snow depth distributions in mountain catchments is critical for applications in hydrology and ecology. Recently, a method was proposed to map snow depth at meter-scale resolution from very-high resolution stereo satellite imagery (e.g., Pléiades) with an accuracy close to 0.5 m. However, the validation was limited to probe measurements and UAV photogrammetry, which sampled a limited fraction of the topographic and snow depth variability. We improve upon this evaluation using accurate maps of the snow depth derived from Airborne Snow Observatory laser scanning measurements in the Tuolumne river basin, USA. We find a good agreement between both datasets over a snow-covered area of 138 km² on a 3 m grid with a positive bias for Pléiades snow depth of 0.08 m, a root-mean-square error of 0.80 m and a normalized median absolute deviation of 0.69 m. Satellite data capture the relationship

between snow depth and elevation at the catchment scale, and also small-scale features like snow drifts and avalanche deposits of a typical scale of tens of meters. The random error at the pixel level is lower on snow-free areas than on snow-covered areas, but it is reduced by a factor of two (NMAD approximately of 0.40 m for snow depth) when averaged to a 36 m grid. We conclude that satellite photogrammetry stands out as a convenient method to estimate the spatial distribution of snow depth in high mountain catchments.

## 1 Introduction

The snow depth or height of the snowpack (symbol: HS, Fierz et al. 2009) is a key variable for both water resource management and avalanche forecasting in mountain regions. However, determination of the spatial distribution of HS in complex terrain remains challenging due to its high spatial variability at horizontal scales below 100 m (Deems et al. 2006, Fassnacht and Deems, 2006). Current approaches to map HS are either based on sparse in situ measurements (Lopez-Moreno et al., 2011, Sturm et al., 2018), area limited unmanned aircraft vehicle (UAV) campaigns (Bühler et al., 2016, De Michele et al., 2016, Harder et al., 2016, Redpath et al., 2018), terrestrial laser scanning (Prokop et al., 2008, Fey et al. 2019) or costly airborne campaigns (Bühler et al., 2015, Dozier et al., 2016, Painter et al., 2016).

Recently a method was introduced to retrieve HS maps from satellite data at metric resolution, typically 1 to 4 m (Marti et al., 2016, McGrath et al., 2019, Shaw et al., 2019). The method is based on the differencing of snow-on (winter) and snow-off (in general end-of-summer) digital elevation models (DEM) that are generated from very high-resolution satellite stereo imagery (e.g. Pléiades, DigitalGlobe/Maxar WorldView-1/2/3 and GeoEye-1). The method was first tested using two Pléiades stereo triplets over the Bassiès catchment in the Pyrenees (Marti et al., 2016). The snow-on and snow-off DEMs were generated using the Ames Stereo Pipeline (ASP, Shean et al., 2016, Beyer et al., 2018) and co-registered before differencing (Berthier et al. 2007). The accuracy of the method was evaluated using 451 probe measurements of snow depth. The HS satellite-derived map was also compared to one obtained from a UAV photogrammetric survey over a small portion of the catchment (3.1 km²). The results showed that snow depth could be retrieved from Pléiades images with an accuracy of roughly ~0.5 m (standard deviation of residuals 0.58 m for a pixel size of 2 m), suggesting that the method had the potential to become a viable alternative to airborne campaigns in mountain catchments with the benefits of a space based platform: access to any point on the globe and lower cost for the end-user. Besides Marti et al. (2016), HS maps from stereo satellite images were evaluated in two recent studies, against terrestrial laser scanning data over a small area (<1 km²) (Shaw et al., 2019) or against ground penetrating radar measurements, which were limited to roughly 50 km² of relatively flat terrain (McGrath et al., 2019).

However, these works provided only a partial validation of the method since the reference data did not homogeneously sample the topographic and HS variability of the study area. For example in Marti et al. (2016), accumulation due to snow drifts on the lee side of high-elevation ridges were not surveyed for safety reasons. The sampling depth was also limited to 3.2 m, which was the length of the snow probes. Furthermore, the areas with steep slopes were under-sampled. Half of the points sampled in the field were on slopes lower than 10° while the terrain median slope in this catchment is ~30°. This lack of validation data in steep slope areas was an important limitation of this study since DEMs from stereoscopic images are known to be less accurate on steep slopes due to a higher sensitivity to horizontal error and to local image

distortion (Lacroix, 2016; Shean et al., 2016). In addition, snow probe measurements may fail to represent the mean HS at the scale of a 2 m pixel especially in mountain terrain (Fassnacht et al., 2018). Furthermore, the impact of the photogrammetric software configuration on the accuracy of the HS map has never been evaluated. The semi-global matching algorithm (Hirschmüller, 2005) was for instance added to the catalogue of algorithms that can be used to derive the disparity map from stereo images in ASP and has never been used with satellite images to derive HS. This algorithm is expected to perform better in low texture terrain (Bühler et al., 2015; Shean et al., 2016, Harder et al., 2016, Beyer et al., 2018) and therefore has the potential to reduce the amount of missing values in the snow depth map.

Given the aforementioned limitations, we present a more comprehensive validation study by taking advantage of the NASA Airborne Snow Observatory (ASO) campaigns in the Sierra Nevada, USA. In this area ASO routinely acquires HS measurements by airborne laser scanning (ALS). We used two Pléiades stereo triplets over the Tuolumne river basin (snow-on and snow-off). The snow-on triplet was acquired on 01 May 2017, the day before the ASO flight and close to the accumulation peak. The ASO product was used as a reference as it should exhibit no bias and was found to have an accuracy roughly an order of magnitude better than Pleiades HS maps (Painter et al., 2016). We use it to test the impact of the DEM processing options, the stereo images acquisition geometry and the HS map resolution on the accuracy of HS. In addition we are able to evaluate an error model (Rolstad et al., 2009), which would enable us to calculate the error of Pléiades HS maps in other study areas where no reference datasets are available.

## 2 Study site

The study site is located in the Tuolumne river basin in the Sierra Nevada mountain range, California, USA (Fig. 1). The Tuolumne river supplies water to the agricultural plain of the Great Valley and the densely populated area of San Francisco. The region recently experienced a five-year drought from 2012 to 2016 (Roche et al., 2018), increasing the interest for water resources monitoring. The ASO flights cover 1100 km² in the basin while this study focuses on a 280 km² subzone which was selected to cover a large elevation range. The elevation within this subzone ranges from 1800 m a.s.l. to 3500 m a.s.l.. Typical winter accumulation can reach several meters at high elevations (Painter et al., 2016). The 2016-2017 winter was characterized by near record snow accumulation that has been referred to as the snowpocalypse (Painter et al., 2017).

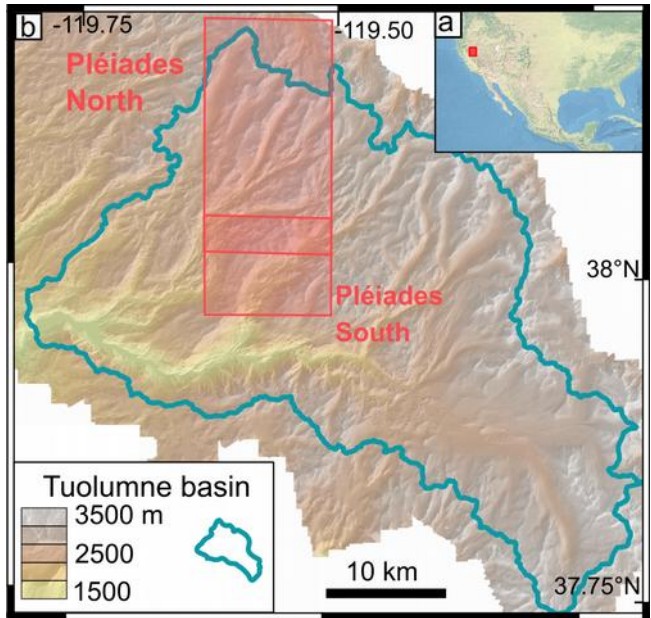

**Figure 1.** The Tuolumne basin is located in California, USA (a). Pléiades images footprint (red polygon) in Tuolumne basin (blue line) (b). The terrain elevation in the background is the snow-off digital elevation terrain from ASO used in the co-registration step.

## 3 Data

 ### 3.1 Pléiades images

The study area is too large to be imaged by Pléiades in tri-stereo mode with a single scene, hence the area was imaged in two strips which overlapped by 3 km in winter and 1.5 km in summer in the along-track direction. The snow-on triplets were both acquired on 01 May 2017, while the snow-off triplets were acquired on 8 August 2017 and 13 August 2017 (Fig. 1, Table 1, images ID in Table S1). The imaged area covers 280 km² in total. Images were acquired in panchromatic and multispectral mode with incidence angles along track between -7° and 9°. The base to height (B/H) ratio of successive pairs is around 0.1 (Table 1). The panchromatic images have a resolution of 0.5 m at nadir and are used to calculate the DEMs. For the snow-on acquisition, we requested to reduce the number of time domain integration (TDI) lines used to image the scene. This is recommended to curb image saturation over sun-exposed snow surfaces (Berthier et al., 2014). As a result, there are no saturated pixel in the images of this study. Pléiades multispectral images have a resolution of 2 m. We only use the multispectral image that was acquired closest to the nadir view angle to compute the multispectral orthoimage. Pléiades images were obtained at no cost for French scientists through the DINAMIS program (https://dinamis.teledetection.fr/). It is also opened to European scientists working in public research institutions. Otherwise Pléiades images can be ordered from Airbus Defense and Space.

### 3.2 ALS data from the Airborne Snow Observatory (ASO)

A snow-off DEM on 13 October 2015 and a snow depth map on 2 May 2017 from the ASO are used for comparison with the Pléiades products (Fig. 1, Table 1). The ASO program, operating since 2012, provides snow depth, Snow Water Equivalent (SWE), and snow albedo maps over full mountain watersheds to support scientific campaigns and operational water management (Painter et al., 2016). The ASO laser scanning system measures the distance between the target and aircraft, and is combined with aircraft position and orientation measurements to generate a collection of reflection points – a "point cloud". Ground points are aggregated to a 3 m grid to derive a gridded DEM (Painter et al., 2015). Snow depth maps are obtained from the difference of a snow-on and snow-off DEM in unforested areas The values on the snow-free areas are used to bias-correct the snow-on elevations and are set to zero. From comparison with 80 in-situ manual measurement, no bias is observed on the HS maps and the root mean square error (RMSE) per pixel at a 3 m resolution is 0.08 m (Painter et al., 2016). For the evaluation of Pléiades HS map, we excluded 25 km² near the catchment divide in the north-east part of the study area because we observed artifacts in the ASO HS map probably due to issues with the aircraft position and orientation data.

**Table 1.** Summary of the data used in this study. The base-to-height ratio (B/H) between the front-nadir (F-N), nadir-back (N-B) and front-back (F-B) pair of images is given for the stereo images.

| Type | Source | Zone | Date | Horizontal resolution | B/H (F-N\|N-B\|F-B) | Snow on/off |
|------|--------|------|------|----------------------|---------------------|-------------|
| Digital terrain model | Airborne laser scanning (ASO) | North+South | 2015-10-13 | 3.0 m | - | Off |
| Snow depth map | Airborne laser scanning (ASO) | North+South (minus 25 km²) | 2017-05-02 | 3.0 m | - | On |
| Tri-stereo images | Satellite optical images (Pléiades) | South | 2017-05-01 | PAN: 0.5 m MS: 2.0 m | 0.12\|0.12\|0.23 | On |
| Tri-stereo images | Satellite optical images (Pléiades) | North | 2017-05-01 | PAN: 0.5 m MS: 2.0 m | 0.12\|0.12\|0.23 | On |
| Tri-stereo images | Satellite optical images (Pléiades) | South | 2017-08-08 | PAN: 0.5 m MS: 2.0 m | 0.12\|0.08\|0.20 | Off |
| Tri-stereo images | Satellite optical images (Pléiades) | North | 2017-08-13 | PAN: 0.5 m MS: 2.0 m | 0.11\|0.11\|0.22 | Off |

## 4. Methods

### 4.1 Workflow for calculation of Pléiades snow depth maps

Figure 2 presents the workflow we developed to produce HS maps from Pléiades images using ASP version 2.6.2 (Shean et al. 2016, Beyer et al., 2018) and the Orfeo Toolbox (Grizonnet et al., 2017). We detail below the calculation of the DEMs, the HS maps and the land cover classifications.

### 4.1.1 DEM calculation

A DEM is computed with the Ames Stereo Pipeline (ASP) using two utilities: *stereo* and *point2dem*. All the options of *point2dem* were set to their default values. We use an iterative approach to obtain a refined point cloud with *stereo* and a DEM with *point2dem* from each triplet of stereo images. The first iteration uses L1B input images to produce a coarse DEM at 50 m resolution. During the second iteration, the L1B input images are orthorectified using this coarse DEM with the ASP utility *mapproject*. The orthorectified images are then processed to obtain a fine DEM at 3 m resolution. The options of the *stereo* command for this second run were empirically adjusted as explained in Sect. 4.1.2. This iterative processing was shown to improve computation time and reduce artifacts in the final DEM (Shean et al., 2016, Beyer et al., 2018). The output DEM resolution and coordinate system was defined to match those of the ASO product (UTM 11 north, WGS 84).

### 4.1.2. Photogrammetric processing of the images

First, *stereo* generates a dense disparity map (i.e.. the pixel displacement between the two images of a stereo pair) using image correlation. The disparity map is used to calculate a point cloud with a triangulation algorithm. Then, *point2dem* interpolates the point cloud on a regular grid (Shean et al., 2016, Beyer et al., 2018). We compared three sets of options in *stereo*. These set of options were empirically selected but do not cover all the options available in ASP. The first set of options is the one used by Marti et al. (2016). This set uses the local-search window stereo algorithm and the normalized cross-correlation parametric cost function with windows of 25x25 pixels (these options hereafter called Local-Search). The sub-pixel refinement algorithm uses an affine method. The two other sets of options use the semi-global matching stereo algorithm (SGM, Hirschmüller, 2005) combined with two different cost functions. The semi-global matching is often used with non-parametric cost functions. Here we compare the two non-parametric cost functions implemented in ASP: the binary census transform (options hereafter called SGM-binary) and the ternary census transform (hereafter called SGM-ternary). The sub-pixel refinement is operated during the SGM correlation with the option *Poly4* of ASP. We evaluated the three sets of options based on the completeness of the maps and the agreement of the snow depth with the ASO using the mean bias, NMAD and RMSE of the residuals. The complete options are available in supplement (Table S2).

The SGM algorithm (Hirschmüller, 2005) differs from local-search window algorithm during the disparity map calculation. The local-search algorithm calculates the disparity for each pixel independently. The SGM algorithm optimizes the disparity over the whole image by assuming that disparity from neighboring pixels is likely to be close. This introduces more continuity in the disparity map and then in the DEM. The matching of subsets of the images of a stereo-pair is measured with a cost function. The binary and ternary census transforms are two cost functions that convert a kernel centered on a pixel into a binary number. For the binary census transform, each pixel of the kernel is compared to the central pixel of the kernel and gives 1 if it is superior to it, 0 otherwise. All the digits are concatenated in a binary number associated with the central pixel. For the ternary census transform, each comparison of a pixel with the central pixel can give three different values: 00,01,11 depending on whether it is smaller, within, or greater than a buffer centered on the central pixel value.

### 4.1.3 Comparison of bi- and tri- stereo images for DEM calculation

We calculated five DEMs from each stereo triplet by selecting a pair of images (front-nadir, nadir-back, front-back) or the complete triplet (front-nadir-back, nadir-front-back). This provided combinations of different B/H (called image geometry further in the article), ranging between 0.08 and 0.23 (Table 1). The three sets of options of *stereo* were tested on these different geometries. In the tri-stereo case, ASP

calculates two disparity maps and performs a joint triangulation when calculating the point-cloud. In the first tri-stereo case (front-nadir-back), ASP calculates a disparity map between the front and the nadir image and between the front and the back image. In the second case (nadir-front-back), ASP calculates a disparity map between the nadir and the front image and between the nadir and the back image. The order of the images matters in the tri-stereo case since the B/H is different between front-nadir and front-back, or nadir-back and front-back. We did not evaluate the third possible tri-stereo combination (back-nadir-front) as we expect results to be similar to the front-nadir-back case.

### 4.1.4 Snow depth (HS) maps

We co-registered the Pléiades DEMs to the ASO snow-off DEM to enable a pixel-wise comparison between both datasets. We first co-registered the Pléiades snow-off DEM to the ASO snow-off DEM. We then separately co-registered the Pléiades snow-on DEM to the Pléiades-registered snow-off DEM before computing the difference between the Pléiades snow-on and Pléiades snow-off DEMs (hereafter referred to as dDEM). The north and south Pléiades dDEMs were mosaiced and the north dDEM value was preserved in the overlapping area. The co-registration vectors were calculated using the algorithm by Nuth and Kääb (2011) on areas where no elevation change is expected (i.e. stable terrain). The stable terrain areas, which are snow free terrain without trees, were determined by a supervised classification of the Pléiades multi-spectral images into a land cover map (see 4.1.5). From the same land cover map, the Pléiades dDEM values were set to zero in snow-free areas to obtain the HS map. Pléiades HS values below -1 m and above 30 m were set to no data to exclude unrealistic outliers based on expert judgment and considering the minimal value that Pléiades HS could reach for actual HS close to zero.

### 4.1.5 Land cover classification

Snow covered areas and stable terrain were analysed separately, and their location determined with a land cover supervised classification calculated from the multi-spectral images. The winter and summer scenes were classified into four categories: snow, forest, open water and stable terrain, the latter corresponding to snow-free areas with low vegetation or bare rock. First, we orthorectified the nadir multi-spectral images using *mapproject* on their corresponding DEM. For each image, we manually extracted training data covering 0.1-1.0 km$^2$ from a composite image of red, green, blue, near-infrared bands and the derived NDVI. A maximum of 33 polygons were manually drawn for the snow class on the winter north image. These samples were used to train a random forest classifier with *otbcli_TrainVectorClassifier* from the Orfeo Toolbox.

The stable terrain and snow masks were shrunk (morphological erosion) with a radius of two pixels (4 m) and patches smaller than 30 pixels (270 m²) were removed. The masks were shifted according to the DEM co-registration vector and then interpolated with the nearest neighbour method onto the ASO grid. Lakes and

225 snow patches remaining in the summer land cover map were removed from the winter snow mask. Lakes were manually delineated on snow-off images. This workflow was automated except for the training dataset which was generated by human interpretation of the images.

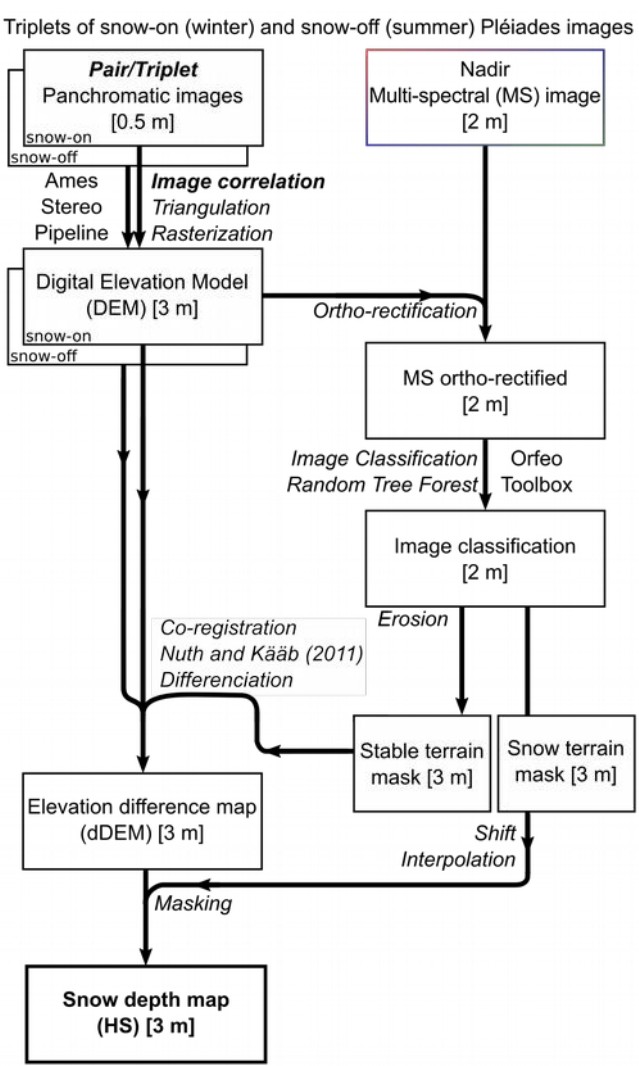

**Figure 2.** Workflow for the processing of the panchromatic and multispectral Pléiades images. Intermediate
products are in the boxes while the processing steps are in italic between the boxes. Text in bold italic characters indicate steps for which we tested different options.

## 4.2 Evaluation of the snow depth maps

We evaluated the Pléiades HS maps for the area where both the Pléiades (snow mask) and ASO (HS greater than zero) HS maps had snow. The term *HS residuals* in the rest of the article refers to the difference between the Pléiades and the ASO HS (Pléiades HS minus ASO HS). We also evaluated the Pléiades dDEM over stable terrain where we expect no elevation difference over time. The stable terrain residuals are the Pléiades dDEMs as ASO products are set to zero over snow-off terrain. The distribution of the residuals was characterized with the bias (both the mean and the median), the root-mean square error (RMSE) and the normalized median absolute deviation (NMAD) of the residuals. The NMAD is a measure of the dispersion suited for populations with outliers (Höhle and Höhle, 2009).

The accuracy of HS maps is often discussed at (or close to) the highest resolution that is allowed by the sensor (e.g. Nolan et al. 2015, Marti et al., 2016). In practice however, HS maps may be subject to spatial averaging to assimilate in a snowpack model, to estimate catchment-scale HS or to compare with coarser satellite products and model output (Painter et al., 2016; Margulis et al., 2019; Shaw et al., 2019). The accuracy of the mean HS of a set of contiguous pixels is expected to be higher than that of a single pixel but depends on the spatial correlation of the errors (Rolstad et al., 2009). Hence, we performed an empirical assessment of the evolution of the accuracy of Pléiades HS as a function of resolution by aggregating the HS residual map to resolutions ranging between 3 m and 180 m (Berthier et al., 2016; Brun et al., 2017; Miles et al., 2018). An average resampling scheme was used, which calculates the average value of all valid contributing pixels. For each resolution, we compared the distribution of the HS residual or measured error to the standard error that is obtained from the error model of Rolstad et al. (2009). Using a spherical semi-variogram model to measure the spatial correlation, Rolstad et al., (2009) estimates the random error of the spatially averaged residual, $\sigma_A$, as:

$$\sigma^2_A = \sigma^2 \left(1 - \frac{L}{l_{cor}} + \frac{1}{5}\frac{L^3}{l_{cor}^3}\right) \quad \text{if } L < l_{cor} \quad (1)$$

$$\sigma^2_A = \frac{\sigma^2 l_{cor}^2}{L^2} \quad \text{if } L > l_{cor} \quad (2)$$

Where $\sigma$ is the standard deviation of the elevation difference, $l_{cor}$ is the semi-variogram range or length of auto-correlation and $L$ is the length of aggregation (half of the pixel spacing). $A$ is the area of aggregation (pixel area) and is related to $L$ ($A = 4*L^2$). This formula assumes no spatial trend in the HS map. We estimated $l_{cor}$ from a semi-variogram analysis of the HS residuals at the highest resolution (3 m). The value of $\sigma$ was taken as the NMAD of the HS residuals at the highest resolution (3 m). We also tested this equation using the stable terrain residuals to set the value of $l_{cor}$ and $\sigma$. The measured error was taken as the

NMAD of the aggregated HS residual. By comparing the measured error and the modeled error, we aim at verifying if i) the error model from Eq. (1) and (2) is valid and ii) its parameters ($\sigma$, $l_{cor}$) can be estimated from stable terrain residuals only. It is important to evaluate if the stable terrain residuals can be used to parameterize the error model because that is the only available information in regions without HS reference data.

## 5 Results

### 5.1 Evaluating the impact of bi or tri-stereo images as input

We first investigate the impact of different image geometries on the HS maps, while keeping the stereo SGM-binary option fixed. The NMAD of the HS residuals with respect to ASO data (Table 2) is larger for maps from pairs of images with B-H around 0.12 (1.13 m for front-nadir, 1.07 m for nadir-back) than from pair of images with B/H around 0.20 (0.68 m for front-back) or triplets of images. The NMAD of the snow depth residuals from the front-nadir-back triplets (0.69 m) is slightly better than from the nadir-front-back triplets (0.78 m) and very similar to the NMAD from the front-back pair. The NMAD over stable terrain is lower but relative values between two geometries are similar (Table 2). For the different image geometries, the RMSE evolves similarly to the NMAD over snow-covered areas but very differently over stable terrain. The largest RMSE over stable terrain is 1.35 m for front-back and the smallest is 1.06 m for nadir-front-back. The mean difference over snow-covered areas ranges from +0.01 m (front-nadir) to +0.16 m (front-back). The absolute means and medians over stable terrain are all less than 0.06 m. The relative results for the different geometries are similar with the SGM-ternary and Local-Search options except for the mean error (not shown here). In the following sections, the HS map from the front-nadir-back geometry is used as it yielded the lowest bias, RMSE and NMAD of all the geometries although similar to the front-back geometry.

### 5.2. Sensitivity to the photogrammetric processing

We compare the *stereo* options on the HS maps from the front-nadir-back geometry (Table 3 and Fig. 3). The SGM sets of options provide DEMs without data gaps. The Local-Search option produces snow-on DEMs with gaps, which results in ~2 km² missing in the HS maps compared to the SGM options (Table 3). Visual examination of the winter DEMs shows large differences in snow fields and forest. Linear artifacts are observed over snow in the DEM produced with the SGM-ternary option (Figure S1). The same regions are noisy in SGM-binary. Patches of typically 20 m x 20 m with abnormally large HS (>10 m) compared to ASO (~3 m) are also observed with the Local-Search options around isolated trees. These artifacts are not visible with the SGM-binary or ternary options (Figure S1).

The mean differences from the ASO snow depth data ranges from +0.08 m (SGM-binary) to +0.49 m (Local-Search option). It is larger for SGM-ternary (+0.24 m) than SGM-binary. The NMAD of the residuals is smaller for SGM-binary (0.68 m) than Local-Search (0.80 m) and SGM-ternary options (0.85 m). Over stable terrain, the absolute mean and median of the elevation differences are less than 0.03 m except for the mean of the Local-Search option which is -0.32 m. The mean of the elevation differences for Local-Search decreases to -0.03 m when the elevation differences are excluded if they exceed three times the NMAD value. This is expected as the same filtering is used during the co-registration process to remove outliers. In the following, the SGM-binary was selected since it gives the lowest bias and NMAD with respect to ASO data and the lowest NMAD over stable areas (Table 3).

## 5.3 Spatial distribution of the residuals

The Pléiades HS map calculated with the selected image geometry and ASP configuration (front-nadir-back images, SGM-binary) compares well with the ASO HS map (Fig. 4). Typical mountain snowpack features (e.g., avalanche deposits and snow drift accumulation) can be identified in the Pléiades HS map (Fig. 4 d.,e., Fig. 5). Pléiades HS data are available over 215 km² of open terrain but not for the 23 km² of forest. No HS was higher than 30 m but 0.25 km² of HS were excluded because HS was less than -1 m. This occurred in areas covered with low density deciduous vegetation which was classified as snow. The intersection area of Pléiades and ASO snow-covered areas is 138 km² after erosion of the Pléiades snow mask. The Pléiades mean (median) HS is 4.05 m (4.13 m) against 3.96 m (4.02 m) for ASO over the common snow-covered area. ASO and Pléiades HS exhibit a similar relationship between HS and elevation (Fig. 6) except between 1900 m - 2100 m and 3500 m - 3700 m where the mean residual over snow-covered areas is greater than 0.25 m (Fig. 7). This corresponds, however, to elevation ranges that cover less than 0.05 km² each.

The NMAD of the Pléiades dDEMs over the 4.07 km² of stable terrain is 0.40 m against 0.69 m for the HS residual. The distribution of residuals on stable terrain is similar for most aspect classes with the exception of the north facing slopes (0.26 km², aspect classes 315°-360° and 0°-45°, Fig. 7). Based on a visual analysis of the residuals map, we attribute these errors to shaded slopes of steep summits. The distribution of HS shows a similar spread for all aspects but a larger positive bias (~0.20 m) for south facing slopes (90°-270°, Fig. 7). The distribution of HS residuals against the terrain slope is similar between 0° and 50°, but has a greater spread in steeper terrain, which covers 2.13 km². The same trend is observed over stable terrain, but only above 70°.

The semi-variogram of the residual increases from 0.2 to 0.8 linearly for lag distances between 3 and 20 m (Fig. 9.a). Low amplitude undulation for lag distances between 2000 m to 8000 m (Fig. 9.b) are related to a low frequency undulation in the HS residual map, which has an amplitude of approximately 0.30 m and a

wavelength of about 4 km (Fig. 8). The crests of the undulation are oriented in the east-west direction (Fig. 8). Such undulation pattern was observed in other Pléiades products, ASTER images (Girod et al., 2017) and World-View DEMs (Fig. 10 in Shean et al., 2016, Fig. 6 in Bessette-Kirton et al., 2018). It is attributed to unmodeled satellite attitude oscillations along-track (jitter). A similar semi-variogram shape is obtained over stable terrain. From this semi-variogram analysis we estimate that the correlation length of the residuals (see 4.2) is about 20 m for both snow and stable areas.

## 5.5 Evaluation of the Rolstad error model

The measured error of the HS map decreases with increasing resampling resolution (Fig. 10). The NMAD of the HS residuals is reduced by a factor of almost two by resampling from the original resolution of 3 m (NMAD=0.69 m) to 36 m (NMAD=0.38 m). As explained in Sect. 4.2, we computed two error models using either the HS residuals ($l_{cor}$= 20 m, $\sigma$ = 0.69 m) or the stable terrain residuals ($l_{cor}$= 20 m, $\sigma$ = 0.40 m) to parameterize Eq. (1) and (2). We find that the NMAD of the HS residuals matches well the error modelled in for averaging areas smaller than $10^3$ m² when $(l_{cor}, \sigma)$ are calculated with the HS residuals (Fig. 10). However it does not match with the modeled error for averaging areas larger than $10^3$ m² (Fig. 10). This is due to the lower decrease of the residuals dispersion with spatial resolution. The measured NMAD decreased by 0.07 m between 36 m resolution and 180 m resolution while the modeled error decreased by 0.22 m between the same resolutions. We attribute this mismatch to the undulation pattern identified in Sect. 5.3 (see Sect. 6.5 in Discussion).

**Table 2.** Comparison of the snow depth residual (HS Pléiades minus HS ASO) and stable terrain elevation difference (Pléiades) for different image acquisition geometries for SGM-binary options only. All metrics are in meters except the mean B/H for bi-stereo geometries, which is dimensionless. The bold line is common to this table and Table 3.

| | Mean B/H | Area (km²) | | Mean | | Median | | NMAD | | RMSE | | Standard deviation | |
|---|---|---|---|---|---|---|---|---|---|---|---|---|---|
| | | snow | stable | snow | stable | snow | stable | snow | stable | snow | stable | snow | stable |
| front-back | 0.22 | 138.11 | 5.2 | 0.16 | -0.03 | 0.18 | 0.01 | 0.68 | 0.39 | 0.80 | 1.35 | 0.79 | 1.35 |
| front-nadir | 0.12 | 138.13 | 5.28 | 0.01 | -0.01 | 0.03 | 0.02 | 1.13 | 0.70 | 1.21 | 1.15 | 1.21 | 1.15 |
| nadir-back | 0.10 | 137.25 | 5.25 | 0.08 | -0.02 | 0.10 | 0.02 | 1.07 | 0.71 | 1.18 | 1.17 | 1.18 | 1.17 |
| **front-nadir-back** | **-** | **138.02** | **5.30** | **0.08** | **-0.01** | **0.10** | **0.02** | **0.69** | **0.40** | **0.80** | **1.16** | **0.79** | **1.16** |
| nadir-front-back | - | 137.51 | 5.29 | 0.13 | -0.06 | 0.15 | -0.00 | 0.78 | 0.44 | 0.90 | 1.06 | 0.89 | 1.06 |

**Table 3.** Comparison of the snow depth residual (HS Pléiades minus HS ASO) and stable terrain elevation difference (Pléiades) depending on the ASP *stereo* options for front-nadir-back geometry only. All metrics are in meters. The bold line is common to this table and Table 2.

| | Area (km²) | | Mean | | Median | | NMAD | | RMSE | | standard deviation | |
|---|---|---|---|---|---|---|---|---|---|---|---|---|
| | snow | stable | snow | stable | snow | stable | snow | stable | snow | stable | snow | stable |
| **SGM-binary** | **138.02** | **5.30** | **0.08** | **-0.01** | **0.10** | **0.02** | **0.69** | **0.40** | **0.80** | **1.16** | **0.79** | **1.16** |
| SGM-ternary | 138.14 | 5.21 | 0.24 | -0.03 | 0.25 | 0.03 | 0.85 | 0.44 | 1.11 | 1.30 | 1.09 | 1.30 |
| Local-search | 135.96 | 5.32 | 0.49 | -0.32 | 0.39 | -0.00 | 0.80 | 0.51 | 1.41 | 1.94 | 1.32 | 1.92 |

## 6 Discussion

### 6.1 Comparison to others studies using satellite photogrammetry

By comparing the Pléiades HS with the ASO data, we find a NMAD of 0.69 m in the best case (i.e. best acquisition geometry and ASP options), which is close to or higher than most previous evaluations (Table 4). Only Marti et al. (2016) measured a larger NMAD (0.78 m) with a reference HS map of 3.15 km² that was obtained by UAV photogrammetry. The spread in accuracy between studies in Tab. 4 could be due to differences in (i) the satellite data (i.e. acquisition geometry, image resolution), (ii) the characteristics of the study site and (iii) the representativeness of the validation data. The comparison with snow probes measurements showed NMAD about a third lower than this study at 0.45 m (n=442, Marti et al., 2016) and 0.47 m (n=36, Eberhard et al., 2020), but covered limited portions of the studied sites. The B/H for the images of Marti et al. (2016) range between 0.21 and 0.25 for all consecutive stereo pairs while our B/H range between 0.08 and 0.12. This is consistent with photogrammetry theory, which states that the accuracy of the DEM increases with the B/H up to a certain limit (Delvit and Michel, 2016). We find a similar NMAD to Eberhard et al. (2020) which calculated a HS map from a Pléiades snow-on DEM and an airplane SfM snow-off DEM and compared it to HS from airplane SfM over 75 km² (NMAD=0.65 m). Finally, McGrath et al., (2019) found a NMAD of 0.24 m for HS from WorldView-3 stereo DEMs using 2107 point observations from ground penetrating radar surveys over a flat area of roughly 50 km². This lower NMAD value might result from the higher resolution of the WorldView-3 images (0.3 m) together with the flatter terrain used for evaluation. As the ASO provides a much larger reference dataset over a complex terrain, we argue that our study provides a more robust evaluation of the HS accuracy that can be expected from Pléiades in high mountain regions. While the ASO data itself may add some error, the published accuracy of the ASO HS data is significantly better than Pléiades. In all these studies, the absolute mean biases range between 0.01 m (McGrath et al., 2019) and 0.35 m (Eberhard et al., 2020).

**Table 4.** Comparison of HS accuracy with studies using satellite photogrammetry. *Eberhard et al. used a Pléiades DEM for snow-on and UAV or airplane SfM DEM for snow-off.

| | Satellite (resolution) | HS map resolution | Validation data | Area | Number of measurements | Mean | Median | NMAD | RMSE |
|---|---|---|---|---|---|---|---|---|---|
| This study | Pléiades (0.5 m) | 3 m | Airplane lidar | 138 km² | | 0.08 | 0.10 | 0.69 | 0.80 |
| Marti et al. 2016 | Pléiades (0.5 m) | 2 m | Snow probing | | 442 | | -0.16 | 0.45 | |
| | | | UAV SfM | 3.15 km² | | -0.06 | -0.14 | 0.78 | |
| McGrath et al. 2019 | WorldView-3 (0.3 m) | 8 m | Ground Penetrating Radar | | 2107 | +0.01 | +0.03 | 0.24 | |
| Shaw et al. 2019 | Pléiades (0.5 m) | 4 m | Terrestrial lidar | 0.74 km² | | -0.10 | -0.22 | 0.36 | 0.52 |
| Eberhard et al.* | Pléiades (0.5 m) | 2 m | Snow probing | | 36 | -0.35 | -0.36 | 0.47 | 0.52 |
| | | | UAV SfM | 4 km² | | -0.18 | -0.18 | 0.38 | 0.44 |
| | | | Airplane SfM | 75 km² | | -0.02 | -0.18 | 0.65 | 0.92 |

## 6.2 Sensitivity to image geometry

We find that the HS maps accuracy is sensitive to the B/H ratio of the input images, and to the configuration details of the photogrammetric processing. We do not find a large added-value of the tri-stereo images for the map accuracy compared to an optimal bi-stereo configuration. Tri-stereo might provide greater benefits in case of image occlusion in steep slopes, which is more prone to occur with higher B/H.

The NMAD of the Pléiades HS is improved by 36 % when using images with a B/H of 0.22 instead of 0.11 (Table 3). Marti et al. (2016) used pairs of front-nadir and nadir-back images (B/H=0.2) as they observed that the front-back pair (B/H=0.4) led to too many no-data pixels. From these two studies and for similar terrain, a triplet of images with a B/H for consecutive images around 0.2 seems a good compromise. It

should ensure high coverage and good DEM precision. Further work is needed to confirm this statement, by testing varying B/H values.

Using tri-stereo instead of bi-stereo images did not improve significantly the Pléiades HS map accuracy. It seems like the processing of a triplet of stereo images (front, nadir, back) with the ASP *stereo* function is equivalent to the processing of the best stereo pair of the triplet, the front-back pair in our case. There were no data gaps due to view obstruction by steep relief in this study area. Should it be the case, the tri-stereo may offer better coverage. Several studies have evaluated the benefits of tri-stereo imagery against bi-stereo

(Berthier et al., 2014; Zhou et al., 2015; Bagnardi et al., 2016; Marti et al., 2016). However, these studies used different photogrammetric software that does not handle the combination of three images in the same way. For example, either multiple disparity maps, or points clouds or DEMs can be calculated and merged to produce a final single DEM. The use of tri-stereo results in increasing the density of the point cloud (Zhou et al., 2015; Bagnardi et al., 2016) and decreasing the area with missing data in the final DEM (Berthier et al.,

2014; Zhou et al., 2015). The accuracy of elevation products from tri-stereo was slightly modified in Berthier et al. (2014) and Marti et al. (2016) compared to bi-stereo, with an increase or decrease of the NMAD by a few percent.

To our knowledge, volume change measurements were never computed from a large number of VHR satellite stereo-images (>10), but studies suggest that the combination of multi-view images can improve the

DEM quality. The fusion of 16 Worldview-3 images improved the NMAD of the residual by 20% compared to a set of 6 images over an industrial zone (Rupnik et al., 2018). Therefore, the most important use of tri-stereo may not be to improve the accuracy of HS maps, but rather to obtain complete coverage of complex terrain and have a less distorted nadir ortho-image for land surface classification. We did not evaluate the extent to which the front and back images would provide a different land surface classification from the one

obtained with the nadir image.

**6.3 Sensitivity to photogrammetric processing**

The choice of the photogrammetric options has an impact on the elevation difference accuracy over stable terrain and snow-covered areas. The NMAD over snow-covered areas is improved by 0.16 m by modifying the cost function (binary census-transform instead of ternary census-transform). However, such a gain on the

dispersion will hardly impact the HS averaged over a region of interest since the random error decreases rapidly with increases in averaging area (see section 6.5). More important is the larger mean bias over snow-covered areas introduced with the SGM-ternary option (0.24 m) and Local-Search option (0.49 m) compared to the SGM-Binary option (0.08 m). This bias is particularly important for south facing slopes. It seems to result from difficulties in image matching in bright areas for the three options and from the impact of

isolated trees for Local-Search. The impact of the trees is likely due to the larger kernel size (25 pixels) used in the Local-Search option. The exact origin of the bias on south facing slopes remains unknown.

## 6.4 Attribution of the HS error

We found a mean difference of +0.08 m between Pléiades (SGM-binary, front-nadir-back) and ASO HS despite the correction of the vertical offset between the snow-on and snow-off DEM using stable terrain after co-registration. This bias is low given the differences in the characteristics of the ASO and the Pléiades products. It can be due to many factors including the effect of vegetation. First, the ASO snow-off DEM is a digital terrain model while the Pléiades snow-off DEM is a digital surface model. Tall vegetation (i.e. trees)

is identified during the classification of the MS images and do not impact the HS evaluation. But short vegetation completely covered with snow in winter is not identified in the classification. For ASO products, filtering based on the multiple lidar returns produced by vegetation should provide the ground elevation, but short vegetation often does not produce multiple returns (Painter et al., 2015). Furthermore, there is a large known error in vegetation height measured with Pléiades DEMs (Piermattei et al., 2018). Thus, it is still

unclear which surface is sensed by each method between the top of the vegetation and the underlying ground.

We found that the random error is larger on snow-covered terrain (NMAD=0.69 m) than on stable terrain (NMAD=0.40 m). This is true for all slopes and most aspects classes (Fig. 7). Although mountainous snow surfaces tend to have smoother topography, thereby increasing the accuracy of the photogrammetric

processing, bright snow surfaces also tend to have less texture than snow-free surfaces, which decreases the accuracy of the photogrammetric processing. The lower accuracy of snow areas is not due to saturation since no pixel was saturated in the panchromatic images. In addition, the residuals over stable terrain were computed from Pléiades data only, while residuals over snow-covered areas were computed from Pléiades and ASO data. Finally, the co-registration of the snow-on DEM was optimized using the stable terrain (Sect.

4.1.4), therefore a lower NMAD on stable terrain may be due to the co-registration step and not the photogrammetric processing in itself. Based on the above, we cannot conclude if the larger dispersion over snow-covered areas results from the properties of the surface.

We further compared Pléiades snow-off DEM with the ASO snow-off DEM and Pléiades snow-on DEM with the ASO snow-on DEM. The latter was calculated by adding the ASO snow-off DEM and the ASO HS.

Both Pléiades DEMs are co-registered as described in 4.1.4. We find a mean bias over snow-covered terrain of +0.13 m for snow-off conditions and +0.21 m for snow-on conditions (Table S3). These biases are of the same order of magnitude and suggest that a bias in the Pléiades snow-on DEM is partially compensated by the difference of the surface observed in the snow-off DEM (see above). In addition, the ASO snow-off DEM was acquired in October 2015 and the Pléiades snow-off DEM in August 2017. Growth or decay of

the vegetation can occur over almost two years, leading to elevation differences between the snow-off DEMs. The NMAD is larger for snow-off DEMs (0.80 m) and snow-on DEMs (0.93 m) compared to HS residual (0.69 m). This shows that some errors are consistently present in the snow-off and snow-on DEMs of each type (airplane lidar or satellite photogrammetry). Pléiades DEMs are indeed over-estimating the surface elevation as the terrain slope increases (Figure S3). This suggests that combining satellite photogrammetry and airplane lidar DEMs may lead to larger errors than comparing DEMs from the same platform.

## 6.5 Evaluation of an error model at different resolutions

The error predicted with Eq. (1) and (2) does not agree with the NMAD of measured HS error for averaging areas larger than $10^3$ m² (Fig. 10). This is likely because Eq. (1) assumes a randomly distributed error beyond the short distance correlation length (here 20 m), while the undulation pattern identified in Fig. 8 introduces an additional spatial correlation at larger scales in the HS residuals map. To verify this explanation, we applied an empirical correction to remove the undulation pattern from the residuals map. We averaged the HS residuals by pixel rows in the across-track direction and used a Fourier transform to identify the undulation frequencies (adapted from Girod et al., 2017). Then, we modelled this error by selecting the frequencies lower than $4 \ 10^{-4}$ m$^{-1}$ (i.e. wavelength longer than 2.5 km) and removed it from the HS map. As expected, this correction makes the semi-variogram of the HS residual flatter for lag distances between 2000 m and 8000 m (Fig. 9.b). As a result, there is a better agreement between the HS residuals NMAD and the modeled error with $\sigma$ and $l_{cor}$ estimated from the HS residuals ($l_{cor}$ = 20 m, $\sigma$ = 0.69 m) (Fig. 10). The improvement is more marked at lower resampling resolution. For instance, the HS NMAD is reduced after correction by 50 % at a resolution of 180 m. The improvement is under 10 % at 20 m resolution as expected since the correction only dampers a low frequency signal. When the stable terrain residuals are used to compute Eq. (1) and (2) ($l_{cor}$ = 20 m, $\sigma$ = 0.40 m), the modeled error is lower than the measured error. This is expected since the NMAD of the stable terrain residuals is lower than the NMAD of HS residual. However, the discrepancy between both models decreased at coarser resolution.

This analysis shows that the model proposed by Rolstad et al. (2009) provides a good first order estimation of the random error after spatial aggregation under the assumption that there is no spatial drift in the error at scales beyond the correlation length. In most cases, the statistics of the HS residuals are not available and might be only measured on stable terrain. Interestingly in this study, the correlation length of the error is similar over stable terrain and snow terrain. However, the dispersion (NMAD, standard deviation) is two folds larger over snow covered terrain than stable terrain, which leads to a proportional underestimation of the error. Finally, although the bias or systematic error is corrected on stable terrain, there remains a bias on

HS of the order of ~0.20 m (Table 4) that should be taken into account in the error calculation. According to the literature, this bias can be estimated by comparing the mean and median of elevation differences over stable terrain (Gardelle et al., 2013) or by calculating the residual of co-registration vector when more than two elevation datasets are available (Nuth and Kääb, 2011).

## 6.6 Comparison of satellite photogrammetry with airborne methods

ALS provides HS maps with a better accuracy (RMSE<0.10 m) than Pléiades and potentially a finer horizontal resolution too (Painter et al., 2016). One significant advantage of ALS is that it can measure HS under the tree canopy and in shaded areas. It is also able to acquire data in overcast conditions provided that the clouds are above the aircraft. However, from this study and Marti et al. (2016), it appears that the accuracy of Pléiades HS maps is sufficient to provide valuable information in regions where there is no ALS monitoring capability (the vast majority of mountain regions with snow cover). A limitation of current very high-resolution sensors such as Pléiades is their narrow swath (20 km for Pléiades) which impedes the acquisition of large areas with a frequent revisit. In particular, there are areas of high tasking competition in lower latitudes where it can be challenging to obtain a stereo pair at the right time of the snow season. More frequent acquisitions should, however, become easier as new stereo satellite fleets are to be launched in the coming years (Pléiades Neo, WorldView legion). The acquisition of visible images will always be limited by the presence of clouds, making some regions hard to study at least during some seasons.

HS maps from UAV SfM typically exhibit a centimetric bias (0.05 m to 0.11 m) and a RMSE between 0.05 m and 0.30 m based on comparison with snow probe and GNSS measurements. This is more accurate than what is currently achieved with satellite photogrammetry. However, UAV campaigns are currently limited to areas of less than 1 km² due to battery limitation and often rely on numerous ground control points. This greatly limits the possibility to cover large and remote areas. Airplane SfM exhibits accuracy close to UAV SfM with NMAD typically of 0.30 m (Bühler et al. 2015) and presents the same potential and logistic limitations as airplane laser scanning campaigns. The reader is referred to the study of Eberhard et al. (2020) for a detailed discussion on the different approaches to map snow depth with photogrammetry.

## 6.7. Generalization to other regions

Several snow applications could benefit from HS maps from satellite photogrammetry. First this study could be reproduced in any place of the globe provided that i) there is a window to acquire snow-off images and ii) there is a way to co-register the series of DEMs, for example with stable terrain. This method is particularly suited for snow volume evaluation at a basin scale in alpine areas (this study site, Marti et al., 2016, McGrath et al. 2019, Shaw et al. 2019). Observing shallow snowpack (roughly HS below 0.5 m, e.g. polar

environments) might not be as straightforward as the typical spatial variability lays within our range of uncertainty (roughly 0.5 m). However, even landscapes with shallow snowpack often feature local accumulation of snow which would be measurable with satellite photogrammetry. Therefore it is hard to qualify this method as unfit to any region, but future studies are required to confirm its usefulness in these challenging contexts. Study of shallow snowpack would clearly benefit from higher accuracy DEMs through correction of the satellite jitter or increases in image resolution.

A lack of well distributed stable terrain in snow-on and snow-off DEMs can complicate the co-registration process in some regions. The horizontal component of the co-registration vector can be measured without differencing stable terrain and snow covered terrain (Marti et al., 2016) but the vertical component requires some stable terrain or an elevation reference. GCPs could be used but would limit the applicability of the method in remote mountains. Besides, it remains to be tested how many GCPs would be required and how precisely their position should be measured.

There are already a number of efficient free and open-access photogrammetric software tools that are under continuous development. These tools enable a high level of automation and are compatible with high performance computing environments (Howat et al., 2019). In our workflow, the last step to automate is the collection of training samples for image classification. This could be done by using an unsupervised classification algorithm or by using an external land cover classification. Preliminary work with a time series of Pléiades images in the Pyrénées (not shown here) suggests that it is not possible to simply use the classification model from a previous year to generate the classification of the current year. A possibility may be to use a Sentinel-2 snow cover map to extract training samples in the Pléiades multi-spectral images, since Sentinel-2 images have a shortwave infrared band which enables a robust and unsupervised detection of snow cover (Gascoin et al., 2019). Differentiating terrain covered with vegetation from stable terrain would remain challenging.

We find that the selection of the image configuration and the processing options can lead to changes in the NMAD up to ~0.3 m. Fig. 10 suggests that this variation is likely to become insignificant if the HS map is aggregated at a larger spatial scale (grids spacing larger than 100 m x 100 m). Such optimisation is therefore more important for the study of small-scale features (wind drift, avalanches, typically about a few tens of meters) or to decrease bias on specific terrain (south slopes, fields with isolated trees). The optimization of the photogrammetric processing can also be important when little stable terrain is available for the co-registration step.

**7 Conclusion**

We found a good agreement between snow depth (HS) maps from high resolution stereo satellite images with airplane laser scanning HS maps over 138 km² of mountainous terrain in California. The mean residual is +0.08 m, the NMAD is 0.69 m and the RMSE is 0.80 m. Comparison of individual DEMs show a growing positive bias with slope in Pléiades DEMs. This bias is of similar magnitude in both snow-on and snow-off Pléiades DEMs and thus cancel out in the HS map, leading to agreement between Pléiades and airplane laser scanning HS for all slopes up to 60°. South facing slopes seem prone to a positive bias in the Pléiades HS (~0.2 m). These areas were found to have less texture in the panchromatic images. The main drawbacks of the satellite stereo HS method are the lack of data under dense tree cover, the reduced accuracy in shaded areas, and the current challenge to image large regions in a short time. We found that the accuracy of the maps was sensitive to the B/H and the photogrammetric processing options. Using the current ASP multi-view triangulation routines, we could not find a clear benefit from the use of a triplet of images compared to a pair with optimal B/H (about 0.2). The accuracy of the HS maps can be improved by decreasing their resolution. This improvement cannot be described with a well-accepted statistical model partly due to an undulation pattern commonly observed in DEMs derived from satellite photogrammetry. We observe that the accuracy is improved by 50 % when decreasing the HS maps resolution from 3 m to 36 m. We conclude that satellite photogrammetric measurements of HS are relevant for snow studies as they offer accuracy of ~0.70 m at 3 m resolution, a high level of automation and the potential to cover remote regions around the world.

*Author contributions*

Conceptualization, CDB, SG, EG, EB; Methodology, CDB, SG, EB, AD; Data Curation and Formal analysis, CDB; Writing – Original Draft Preparation, CDB, SG, EB; Writing – Review & Editing, CDB, SG, EB, DS, AD, JD, EG, MD; Supervision, SG, EB, MD; Funding Acquisition, SG, EB, MD, CDB.

*Acknowledgements.*

This work has been supported by the CNES Tosca and the Programme National de Télédétection Spatiale (PNTS, http://www.insu.cnrs.fr/pnts), grant n°PNTS-2018-4. The National Center for Atmospheric Research is sponsored by the US National Science Foundation under Cooperative Agreement No. 1852977. Additional support provided by a cooperative agreement with the US Bureau of Reclamation Science and Technology Program. The authors would like to thank Thomas Shaw and Renaud Marti for their careful read of the manuscript.

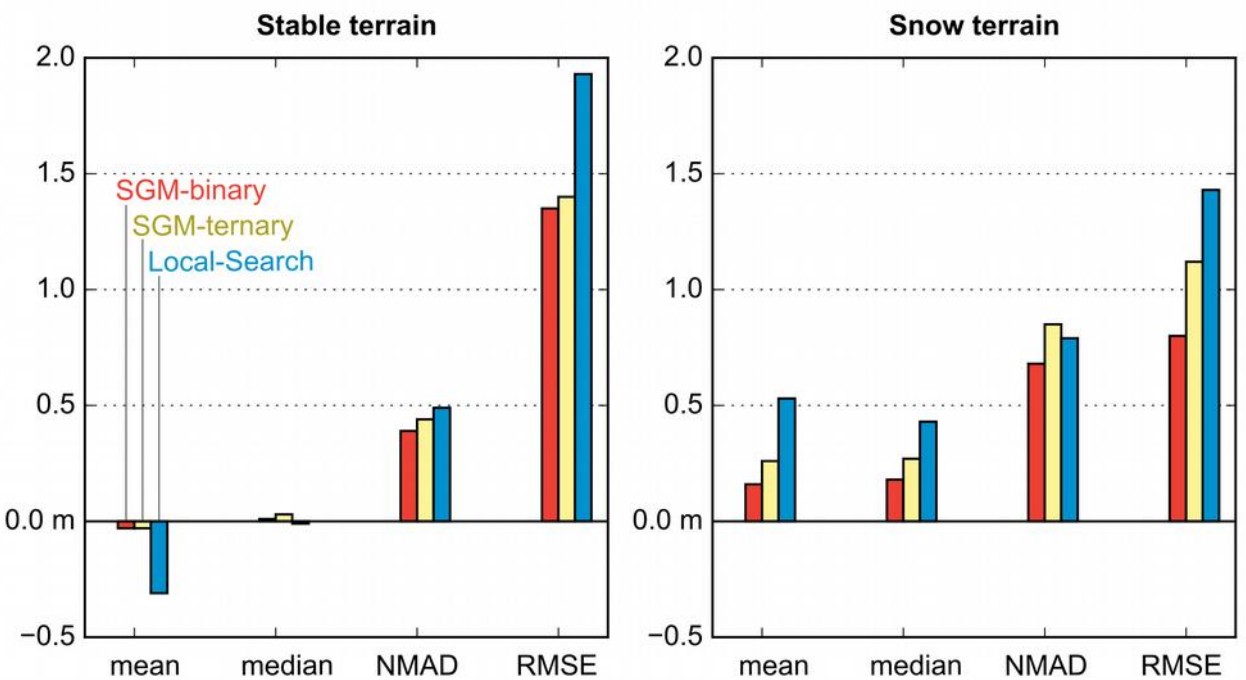

**Figure 3:** Mean, median, NMAD and RMSE of the residual of HS maps depending on the ASP *stereo* correlation option. The options compared are the SGM algorithm with the binary census-transform cost function (SGM-binary in red), with the ternary census-transform cost function (SGM-ternary in yellow) and the local search algorithm (Local-Search in blue).

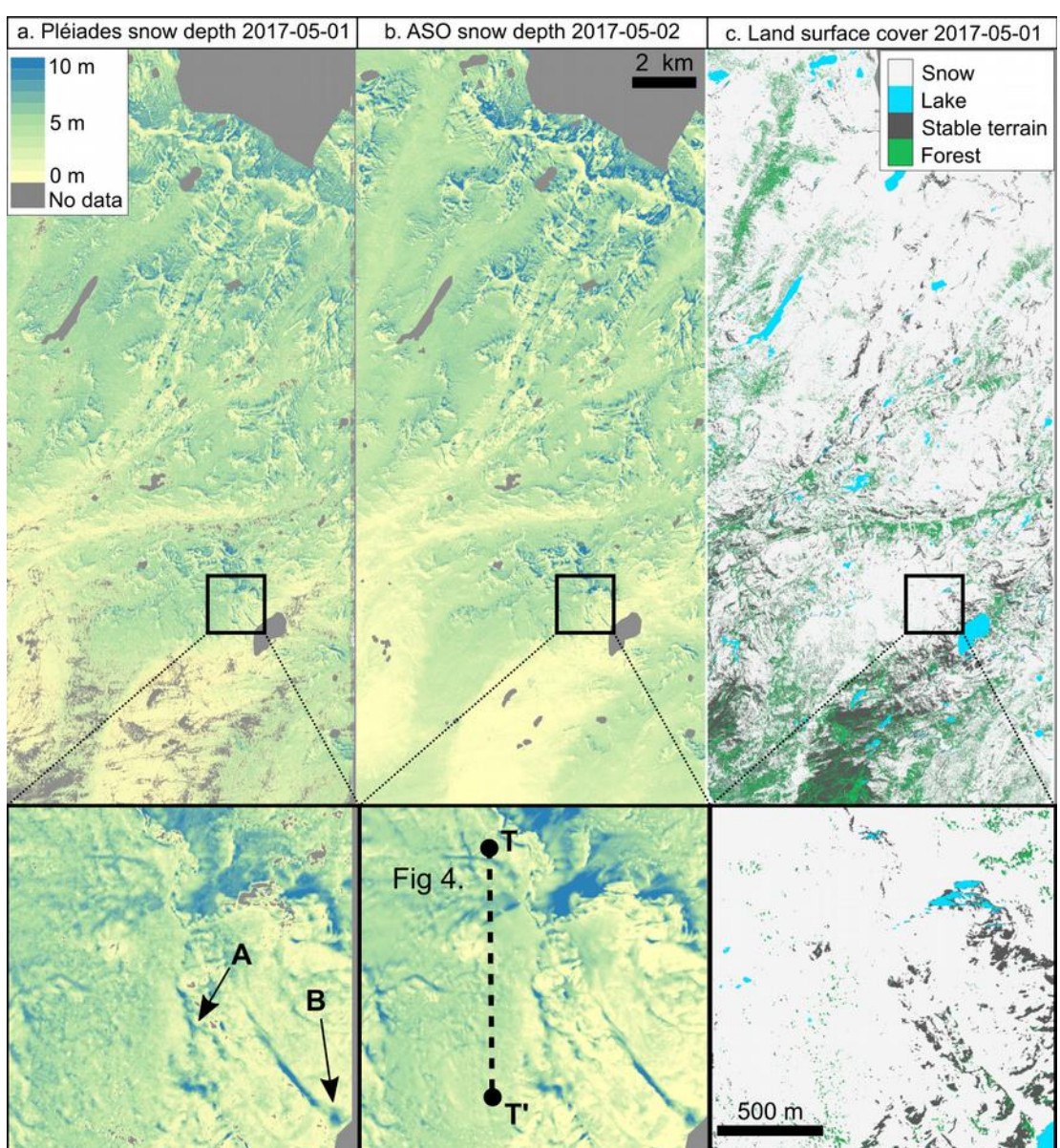

**Figure 4**: Snow depth maps from Pléiades data on 01 May 2017 (a and d) and from ASO on 02 May 2017 (b and e). Cornice (A) and avalanche deposits (B) are visible on Pléiades HS maps (d). The land surface cover is shown in c and f over the same area. Black squares in a, b, c, is the area shown in d, e, f. The transect T-T' is shown in Fig. 5. All datasets have the same spatial resolution (3 m). The difference of the maps (a) and (b) (Pléiades minus ASO) is in Fig. 8.

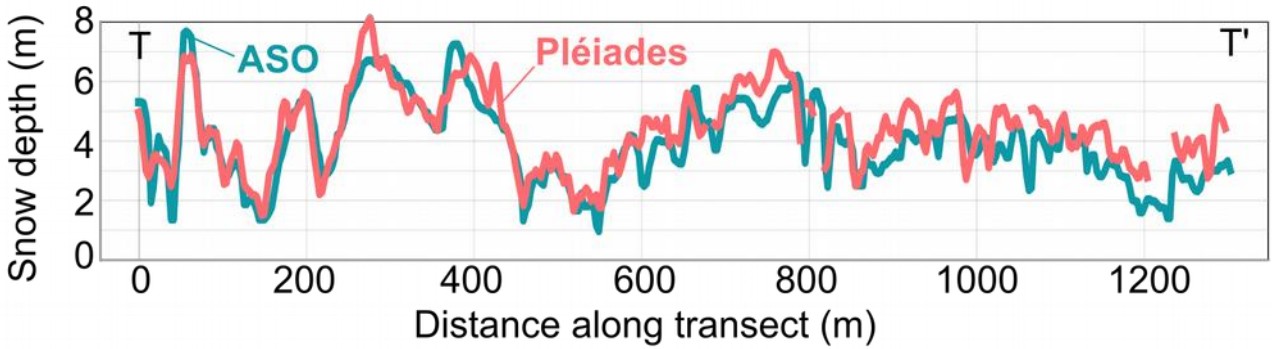

**Figure 5.** Transect T-T' of snow depth visible on Figure 4. e. from Pléiades data (pink) and ASO (blue). More transects are available in supplement.

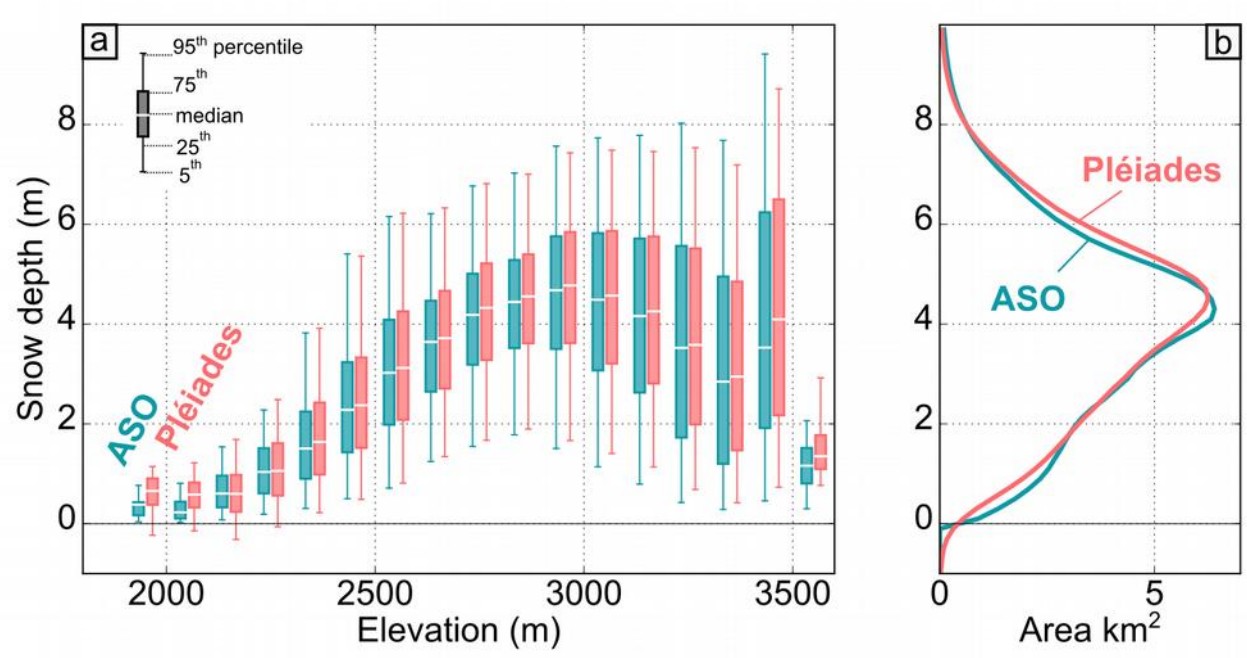

**Figure 6.** Snow depth against elevation (a) and total distribution (b) from Pléiades data (pink) and ASO (blue). The boxplots show the median value (white line), the 25th and 75th percentile (box) and the 5th and 95th percentile (whiskers).

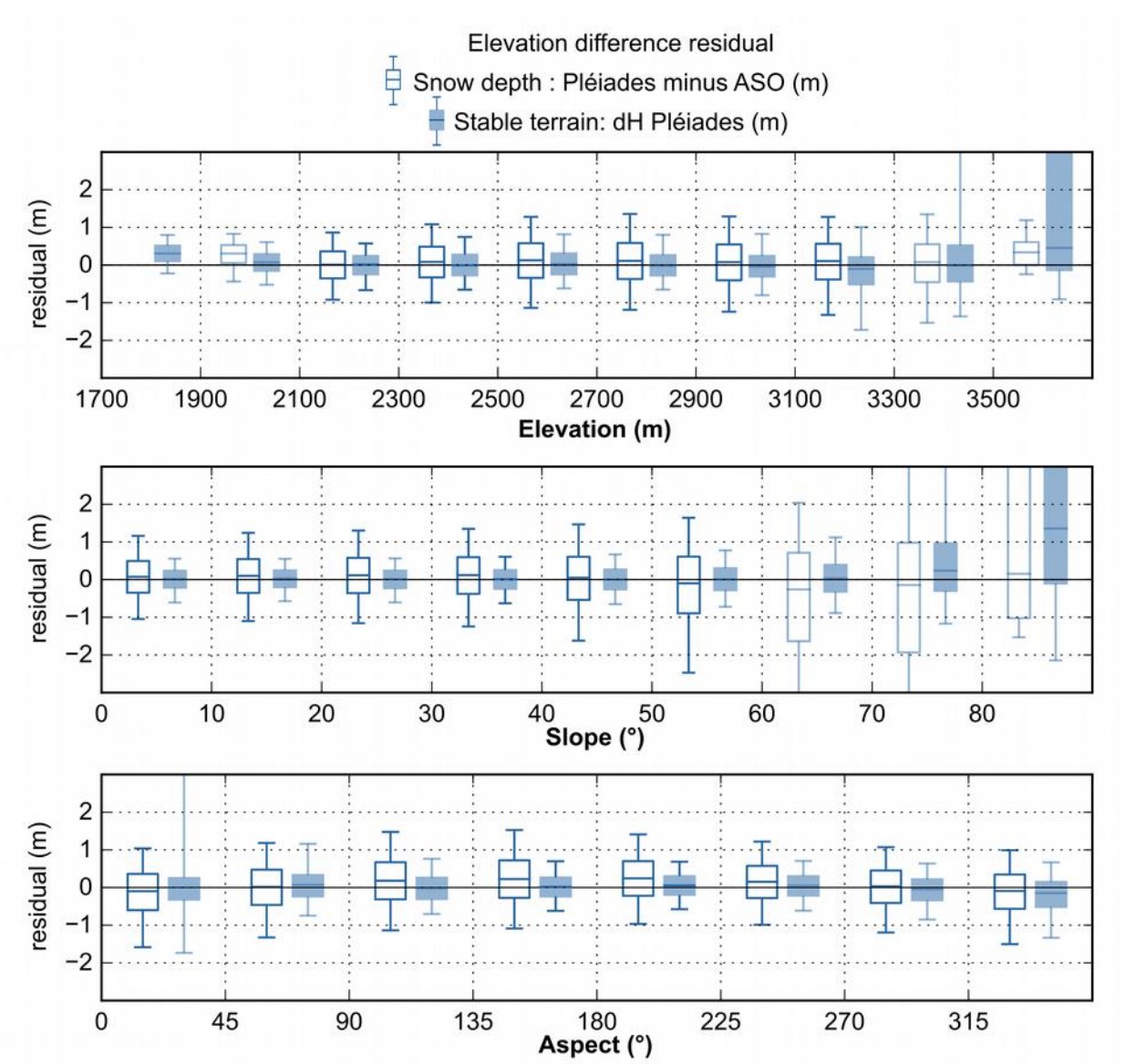

**Figure 7.** Distribution of the residuals between the Pléiades and ASO snow depth maps over the snow-covered area (empty box) and stable terrain (filled box) against elevations (top), slopes (middle) and aspect (bottom). Over stable terrain, ASO product is set uniformly to zero. Boxes where data were covering less than 1 km² are slightly transparent.

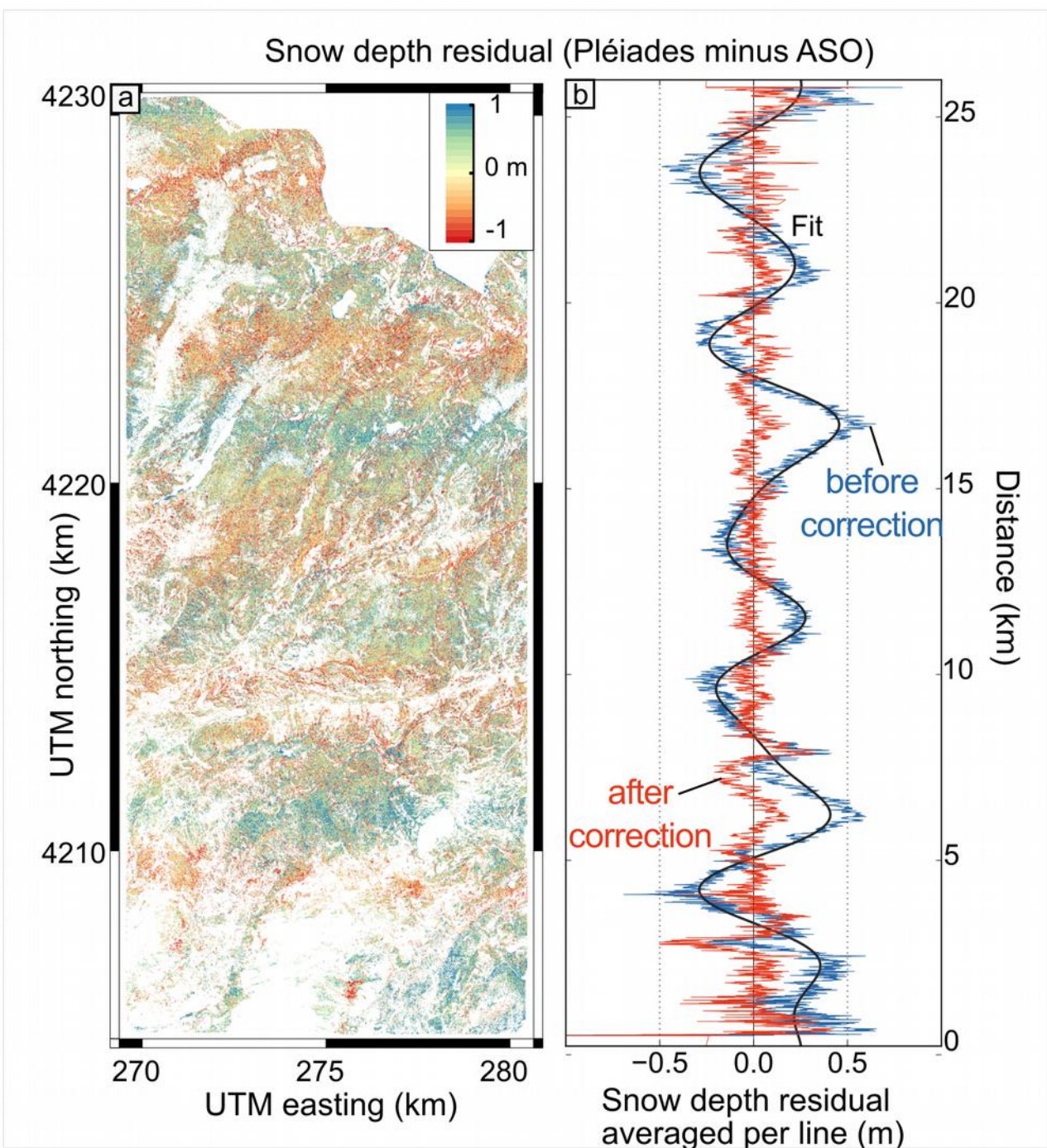

**Figure 8.** Residual snow depth (Pléiades minus ASO) over the complete study area (a) and average per line

(b). In b., the HS residual before correction (blue) is corrected for the low frequency undulation (black) to

obtain a corrected signal (red).

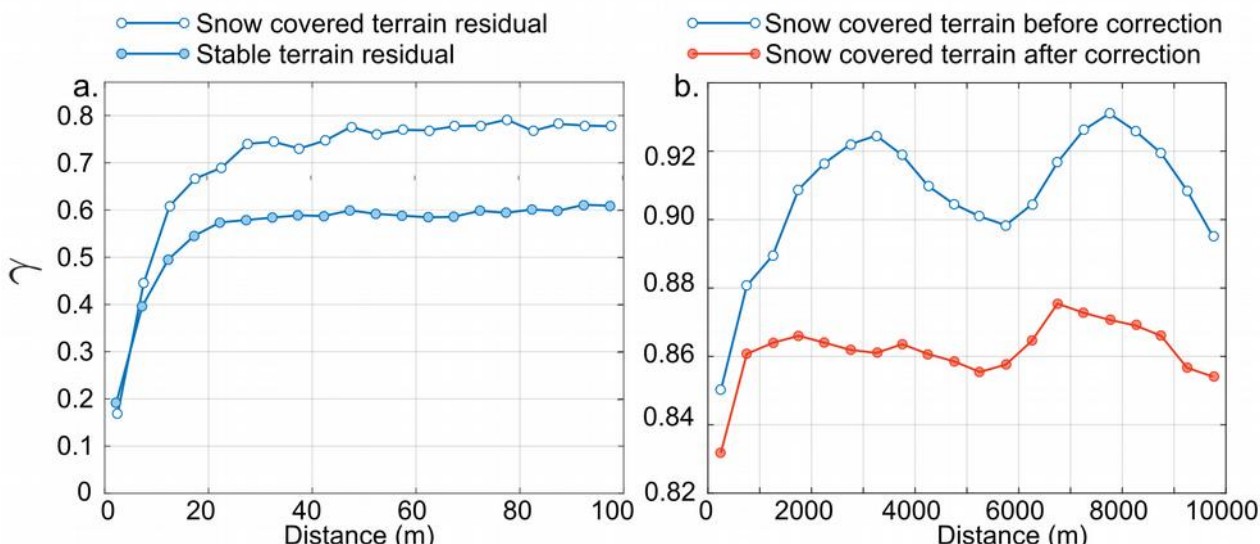

Figure 9. (a) Semi-variogram or spatial autocorrelation (γ) against lag distance of the HS residuals (empty circles) and Pléiades elevation difference over stable terrain (filled circle). (b) Semi-variogram of the HS residuals for large distances before (blue line) and after correcting the undulation pattern (red line), illustrating the reduction in spatial variance at greater lag distances due to this correction (Sect. 6.5).

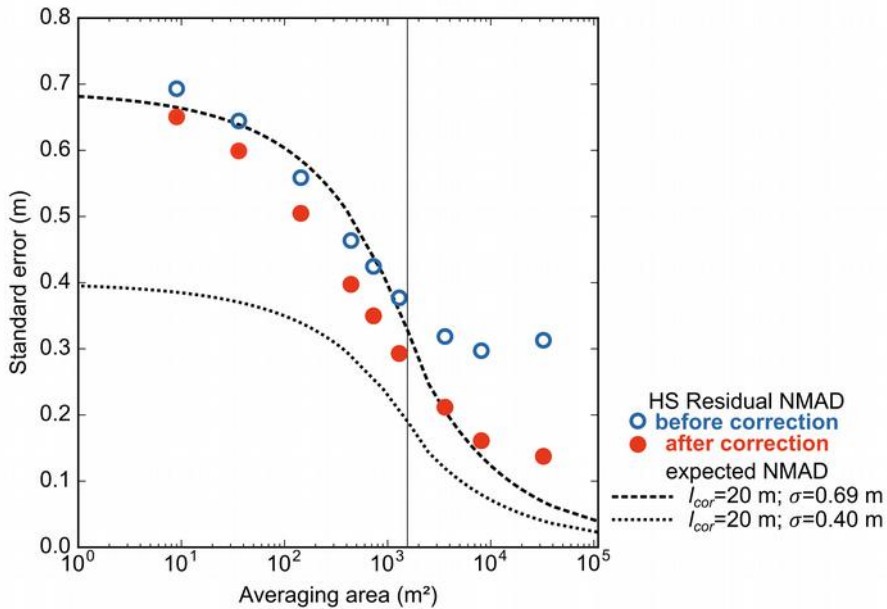

Figure 10. Measured error and modeled error (Rolstad et al., 2009) of the HS averaged over different areas. Modeled error (dashed line) is predicted based on different random error per pixel ($\sigma$) and autocorrelation length ($l_{cor}$) (Eq. 3). The lines are the modeled error based on $l_{cor}$ from the semi-variogram and $\sigma$ taken as the NMAD derived from stable terrain residuals (dotted line) and HS residuals (dashed line). Empty blue circles are the NMAD of the residual HS maps averaged at different resolutions before the undulation correction.

Filled red circles are the NMAD of the residual HS maps averaged at different resolutions after the undulation correction.

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
