# Peer review of "Snow depth mapping from stereo satellite imagery in mountainous terrain: evaluation using airborne laser scanning data"

_The Cryosphere, 2020_

## Referee Comment (RC1) · Yves Bühler (Referee) · 25 Feb 2020

The paper entitled "Snow depth mapping from stereo satellite imagery in mountainous terrain: evaluation using airborne lidar data" by C. Deschamps-Berger et al. presents the second sound evaluation of VHR satellite data for snow depth mapping in alpine terrain after Marti et al. 2016. In their study, they apply high-quality, spatial continuous LiDAR as reference dataset, allowing for a meaningful validation over a large region. This is an important and valuable contribution to further advance snow depth mapping based on remote sensing data. However, there are four main points I would like to see clarified and complemented before I can recommend the paper for publication:

[Figure]

1. Error combination of snow-on and snow-off DSMs:

In their study they compare HS maps form Pléiadessnow-on minus Pléiadessnow-off with HS maps generated from LiDARsnow-on minus LiDARsnow-off. Certainly, this comparison makes sense and provides interesting insights but only as a second step. First, we need to see how much error is coming from individual surface models. If you take the snow-off and the snow-on DSM from the same platform and processing the errors of the individual DSMs could a) be added to each other or b) erase each other. So, if the snow-on DSM has a high error on one side (e.g. being too high) and the snow-off DSM has the error to the same side, the resulting snow depth is accurate even though both DSMs are bad. In the comparison as it is done now in the paper, there is now chance to see the quality of the individual DSMs. Therefore, I request the authors to do a HS comparison applying also the LiDARsnow-off DSM for the Pléiadessnow-on DSM. This will reveal how the snow-on DSM performs excluding random error addition from the snow-off DSM. Also, a DSM comparison between the Pléiadessnow-off and the LiDARsnow-off DSM should be performed and discussed.

2. Justification of selected photogrammetric processing method:

The authors select three different processing methods in ASP (SGM-binary, SGM-ternary and Local search). But they do not justify why they selected these algorithms nor do they describe the algorithms and their differences. Are there other options? An overview on the available algorithms including their strength and weaknesses would be very helpful for the readers and would allow for the justification of the selection. Why does the SGM binary perform best? This section has to be expanded.

3. Comparison to results from other photogrammetric platforms (UAS and airplanes) and discussion of photogrammetry specific issues:

The embedding of the work into the current literature is weak (section 6.1). It is true that they are among the first mapping snow depth with optical satellite data. But there has been a lot of additional work concerning photogrammetric snow depth mapping
from UAS platforms (e.g. Van der Jagt et al. 2015, Bühler et al. 2016, De Michele et al. 2016, Harder et al. 2016, Adams et al. 2018). The authors should compare their results also to these publications and discuss more about the specific problems of photogrammetry on homogenous snow-covered areas (e.g. Bühler et al. 2017) and the effect of vegetation. I assume that there is distinct problem with noisy surfaces. This topic is not sufficiently discussed in the paper.

4. Discussion on which applications can benefit from this approach and which not:

With the elaborated accuracies in mind, there should be a discussion on which applications could benefit from this new method and which not. For example, regions with shallow snow cover (0 – 50 cm) over the vast regions on the higher latitudes of the northern hemisphere would be within the error range. So, I assume the method would be not accurate enough. But for what applications could it be of big value? I think this would be a very interesting part for the readers.

Detailed comments:

P3L65: Why are DSMs less accurate in steep terrain? Is it mainly because a small x,y shift results in a large z-shift? Please explain a bit more

P3L76: the term airborne lidar altimetry sound a bit strange to me, I would use airborne laser scanning ALS.

P3L87: Please give some references for this statement

P4L103: Why did you select exactly this zone? Can you give some justification?

P4L118: How was the image acquisition performed? What prices did you have to pay? What would be the conditions for people who want to do the same as you did?

P5L135: Why is the resolution 3 m is there a justification for that? What was the point density per m2 of the LiDAR dataset? Double period at the end of the sentence.

P7L173: In my experience it can be very hard to find snow free, stable terrain in

high alpine regions and this with a good distribution over the entire investigation area. Please discuss this issue and give some recommendations what to do if not enough stable terrain can be found. Please also discuss why an absolute orientation via GCPs is not suitable.

P7L176: Why did you choose the thresholds -1 to 30 m? Please justify

P7L182: In my experience a big problem are alpine bushes (0.5 to 2 m high). How do you treat these? It is not mentioned

P7L190: Why do you use nearest neighbor and not cubic resampling? Please justify.

P10L244: Artifacts in the equations, please check all equations

P11L266: It would be very helpful for the readers to have a figure showing the matching success and the hillshade of the DSMs generated with the different pairs This would also clarify the artifacts you mention.

P12L296: Why do you use the threshold 3 x NAMD? Please justify.

P13L325: Do you have a hypothesis where this error is coming from?

P13Table2: There is a 10 cm difference in RMSE and NMAD between front-nadir-back and nadir-front-back. Why is this showing, please explain as you take basically the same input data just in reversed order.

P14L346: Please extend this section substantially by discussing results from previous UAS and airborne investigations including the reached accuracies and performance on snow covered surfaces (see my main point 3)

P14L362: But please mention the benefit of additional coverage in particular within steep slopes.

P14L366: Can you make the statement "B/H around 0.2 seems to be beneficial" stronger by providing more justification

P15L400: what was calculated ASO minus Pléiades or the other way round? ASO should be higher as it uses a DTM as snow-free DSM

P17L449: Please expand this section taking into account published UAS and airborne photogrammetric investigations

P18L473: The conclusions are a bit thin. Please also provide the RMES values. Could you provide more important information? The term "good accuracy" is vague.

P20Fig4: The scale bar is too large hindering a sound interpretation of the values. I would propose a range of 0 to >5 m. Here a difference image (HS_ASO minus HS_PLéiades would be essential. This would allow for a detailed interpretation of the results, also for the subsets. What you name an avalanche deposit in d) does not look like that to me at all. I saw many avalanche deposits in HS maps and I am pretty sure that this one is something else. Please check that. Why did you locate the transect there? Can you give a justification?

P21Fig5: Here it would help if you could draw some more transects at further locations to see the differences

P21Fig6: The ASO HS median is mostly lower than the Pléiade HS even though the ASO uses a DTM as snow-free surface (penetrating low vegetation). Therefore, we would expect the LiDAR HS to be slightly higher. Can you comment on that?  

References:

Adams, M. S., Y. Bühler, and R. Fromm (2018), Multitemporal Accuracy and Precision Assessment of Unmanned Aerial System Photogrammetry for Slope-Scale Snow Depth Maps in Alpine Terrain, Pure and Applied Geophysics, 175(9), 3303-3324, doi:10.1007/s00024-017-1748-y.

Bühler, Y., M. S. Adams, R. Bösch, and A. Stoffel (2016), Mapping snow depth in alpine terrain with unmanned aerial systems (UASs): potential and limitations, The Cryosphere, 10(3), 1075-1088, doi:10.5194/tc-10-1075-2016.

Bühler, Y., M. S. Adams, A. Stoffel, and R. Boesch (2017), Photogrammetric reconstruction of homogenous snow surfaces in alpine terrain applying near-infrared UAS imagery, International Journal of Remote Sensing, 8-10, 3135-3158, doi:10.1080/01431161.2016.1275060.

De Michele, C., F. Avanzi, D. Passoni, R. Barzaghi, L. Pinto, P. Dosso, A. Ghezzi, R. Gianatti, and G. Della Vedova (2016), Using a fixed-wing UAS to map snow depth distribution: an evaluation at peak accumulation, The Cryosphere, 10(2), 511-522, doi:10.5194/tc-10-511-2016.

Harder, P., M. Schirmer, J. Pomeroy, and W. Helgason (2016), Accuracy of snow depth estimation in mountain and prairie environments by an unmanned aerial vehicle, The Cryosphere, 10(6), 2559-2571, doi:10.5194/tc-10-2559-2016.

Van der Jagt, B., A. Lucieer, L. Wallace, D. Turner, and M. Durand (2015), Snow Depth Retrieval with UAS Using Photogrammetric Techniques, Geosciences, 5(3), 264-285, doi:10.3390/geosciences5030264.
* * *

---

## Referee Comment (RC2) · Phillip Harder (Referee) · 9 Mar 2020

Review of Deschamps-Berger et al. "Snow Depth Mapping from stereo satellite imagery in mountainous terrain- evaluation using airborne lidar data

This work provides a deeper evaluation of the snow depth mapping from stereo satellite imagery approach proposed by Marti et al (2016). The advancement of this work is to consider some of the various/emerging DSM processing options and a more thorough error analysis with basin scale airborne lidar data from ASO versus manual discrete manual probe observations in locations not fully representing the variability present in the landscape. The prospect of obtaining high resolution (3m scale here) snow

depth with errors reported here to be less than 0.8m from a space borne platform is tremendously exciting and the benefits of such a capability are clearly articulated herein better than I can summarise in this space.

Much of this article is clear and well written but there are a couple aspects which would benefit from some clarification and/or clearer justifications. I will begin with some main comments and then provide a list of more technical comments/edits/suggestions. Overall I think this work is well suited to The Cryosphere. The previous review of Buhler makes many important observations which I fully agree with. I would highly recommend the authors make those edits in addition to some more articulated here.

Main Comments:

1) Error model: there is a lot of discussion of methodology and results of the scaling of the random error with length scale. I have a couple concerns on this. First we are looking at the random error metric, articulated later on as the standard deviation of the snow depth residual error. This is only one part of the error, as in overall error is comprised of random components (captured here) as well as biases (not a part of this model). Correct me if I am wrong but my read is that increasing length scales will lead to decreases in random error and will therefore not comment on the bias error? Or, this model shows that increasing the length scale will increase precision but does not say anything about the accuracy? We could have really large biases in the dDEM but these will not be reflected in the model? A quick read from the abstract doesn't articulate this nuance that I am perceiving. Without lidar or ground observations to correct for the bias this suggests there will be operational challenges to implement this method in data sparse regions. Would co-registration on common snow-free stable areas be a reasonable approach to provide these relative differences? Would you have any other suggestions to improve applicability in data sparse areas? Second, there is an assumption that stable terrain residual random errors apply to snow surfaces. Is this a valid assumption? In most topography the snow surface will reflect the underlying ground surface. In complex alpine topography the range in surface elevations can be
orders of magnitude greater than the hs corresponding to the dDEMs and therefore this is a reasonable assumption. But in many areas prone to wind redistribution parts of the landscape can be smoothed out, for example, wind blown snow fills up gullies. Therefore, I would expect the random components of the error to vary between snow and snow-free area. While the ability to predict error of a snow depth product is of interest I worry that this is perhaps a little too simplistic and detracts from the main point of the paper which is a detailed evaluation of satellite stereo imagery snow depth measurement and its comparison to airborne lidar data. The comments from Buhler about partitioning the error the errors of the snow-free and snow covered surfaces would be very valuable and provide a clearer interpretation of what the HS error is derived from.

2) Structure: There is a lot of detailed technical discussions but many sections of this paper would benefit from stepping back for am moment and explaining the justification for what is occurring and how it fits in to the overall story. As it is there are some disjointed sections. (2 examples: line 125-126 – this sentence requires context, line 240 onwards – why do we care about developing an error model?)

Specific comments:

Line 21 "A recent new method" -> "Recently, a method"

Line 32: "up to" -> "down to"? I don't see anywhere else in the paper where it is mentioned that there is a factor of two decrease in random error going from 3m to 20m (short of a reader interpretation of Figure 10). Also as a sample size increases the standard deviation (aka random error here) will always decrease – can you articulate the nuance that the modelling modifies this relationship by accounting for spatial correlation?

Line 35-37: A great conclusion!

Line 44: Nolan et al. 2015 is not a UAV reference, they use SfM from a manned

airplane. Other early reference options would be Buhler et al. 2016, Harder et al. 2016 or De Michele et al. 2016 for starters.

Line53-55: One criticism from abstract was that the Marti paper only considered limited numbers of observations and terrain representation with manual insitu probe depths while here there is acknowledgment that there was also validation versus UAV data? Clarify this contradiction?

Line 72: Harder et al. 2016 also showed that SGM improved performance over low texture snow in the UAV-SfM snow depth mapping context.

Line 136-137: this is not a method to do forest snow depth mapping so is it important to include forest lidar data processing steps?

Line 162: if including coordinate system information should also put in the projection.

Line 175: What was the extent of the Pleiades HS values that was outside these thresholds?

Line 188: Is "eroded" the proper term to describe this? I feel this may be confusing for those with geomorphology mindsets.

Figure 2: Line passes through the "co-registration" step. Does this need a box or to be offset like other processing step descriptions?

Line 240 -255: empty super and sub script boxes appear in many of the equation/symbol text. Add space between equations and the equation numbers as well?

Section 5.1. Can you clarify the results and discussion around the comparison of pairs versus triplets? In parts (like the last sentence of this section you say the triplet is the best) while in others (line 360) it is justified that a pair of images is just fine. Line 382-384 says tri-stereo is best to provide the best coverage and reduce distortion. Can this be clarified to allow for more consistency throughout?

Line 320-325: the low frequency undulation in HS residuals. Where is this coming

from? How pervasive is this error with satellite stereoscopy or is it specific to this site/processing options? For this approach to be of value what techniques can be employed to address this error (while low amplitude could be important) where there is no ASO like lidar data for validation?

Line 330-334: are you applying the error model to the undulation removed HS residual or the raw HS residual? Can you clarify that? If you consider a semivariogram that extends out to the amplitude of the undulation does the undulation length scale appear in the semirvariogram (extending figure 9)

Line 367-387: there is a justification being made here to bi-stereo imagery versus tristereo. Can you also articulate/quantify what the differences there may be in terms of cost differences (financial and computing). Will help to justify from a resource perspective why we should consider doing this all with pairs of images if it can articulated that there are significant savings terms of money and computing time/power requirements.

Line 410: can you clarify "decimetric accuracy"?

Line 421-424: merge with following paragraph?

Line 430+: is this supposed to be a new paragraph?

Line 436: "squares of length 210m". squares themselves don't have a length – can this be expressed differently?

Line 438: "unvalidated" -> "invalidated"?

Line 455: "probably the vast majority of mountain regions with seasonal snow cover" -> "the vast majority of mountain regions with snow cover"

Line 458-459: "high competitions especially" -> "high tasking competition"

Line 459-260: can you identify specific satellite platforms that we should keep a look out for?

Line 471: this is an important point to make that needs more than 1 sentence at the end of the discussion. Can you emphasise the implications of this on where snow-depth mapping with this technique is valid and possible steps that may be available to address this limitation?

Line 474-475: "satellite very high resolution stereo images" -> "high resolution stereo satellite images"

Figures: Information in the figures is good but the layout of the figure themselves need a fair bit of formatting work.

Figure 3, 5, 6, 7, 8, 9 , 10: Can you pull out the text/lines from inside the plotting areas in to legends (based on color) outside .

Figure 4: all of the lettering (referencing inset maps etc. ) is a little confusing in the legend. Perhaps add a upper-lower case distinction?

Figure 3, 7: Put x-axis descriptions outside of the plotting areas

Figure 7: slope residual plot quantiles exceed the plot area.

Figure 8: could the a? panel have coordinates (UTM?) to provide scale and could then remove the scale bar. Also remove "Map" title and on b remove the titles and add x-axis label and units.

Figure 9: "distance" -> "Distance"

Figure 10: can you provide an explanation for "raw and "corrected" in the figure caption?

References:

Bühler, Y., Adams, M. S., Bösch, R., Sto, A., Buhler, Y., Adams, M. S., Bosch, R. and Stoffel, A.: Mapping snow depth in alpine terrain with unmanned aerial systems (UASs): Potential and limitations, Cryosphere, 10, 1075–1088, doi:10.5194/tc-

10-1075-2016, 2016.

Harder, P., Schirmer, M., Pomeroy, J. W. and Helgason, W. D.: Accuracy of snow depth estimation in mountain and prairie environments by an unmanned aerial vehicle, Cryosph., 10, 2559–2571, doi:10.5194/tc-10-2559-2016, 2016.

De Michele, C., Avanzi, F., Passoni, D., Barzaghi, R., Pinto, L., Dosso, P., Ghezzi, A., Gianatti, R. and Vedova, G. Della: Using a fixed-wing UAS to map snow depth distribution: An evaluation at peak accumulation, Cryosphere, 10(2), 511–522, doi:10.5194/tc-10-511-2016, 2016.

---

## Referee Comment (RC3) · Steven Fassnacht (Referee) · 19 Mar 2020

This is an important paper as it presents the implementation of relatively new technology, stereo satellite imagery, to map snow. It provides the first comprehensive evaluation of a snow depth dataset derived from stereo satellite imagery by comparing it to an extensive lidar dataset (ASO). Overall the paper is presents important step towards better snow mapping.

The science is good and meaningful. The writing is not easy to read in many places. Specifically, the text could be more concise, and I recommend that the authors revisit how the paragraphs are structured and how sentences are written. While the English

is not incorrect, the flow of the sentences makes it difficult to read. I acknowledge that the main authors are not native English speakers, but some of the 8 authors are native English speakers, and most of the authors have written extensively in English. I will use the Introduction as an example. It provides useful information, but is awkwardly written. The material reads partly like a stream of consciousness. There are new paragraphs that are a continuation of the previous paragraph. Please consider restructuring this section. For example, the text starting on lines 49-50 through line 73 presents the application of the stereoscopy using satellite imagery. Yet it starts in the middle of a paragraph with "The method was tested using two Pléiades stereo triplets over the Bassiès catchment in the Pyrenees (14.5 km$^2$)." Then the authors tell us about what was done there. The next paragraph begins with "However" and is a continuation of the first paragraph. The last paragraph does present additional steps that will be seen in the rest of the paper. In restructuring the Introduction, end with objectives addressed or research questions posed so the reader knows where this paper is going.

There are various terms used that, while not incorrect, are awkward, such as on line 24 "deepen" in the context of a limited evaluation or on line 42 "decametric scale" to talk about variability over 10s of meters. Similarly, some of the phrasing can be more succinct. For example, on lines 232-233 the authors state: "We evaluated the quality of the Pléiades HS maps over the area defined as the intersection of snow-covered terrain in Pléiades HS maps (snow mask) and ASO HS maps (HS greater than zero)." How about: "We evaluated the Pléiades HS maps for the area where both the Pléiades (snow mask) and ASO (HS greater than zero) HS maps had snow. This is stylistic, but some of the text is more complicated than it needs to be, and thus makes the paper more difficult to read.

Specific comments:

- Lines 77-78: "The snow-on Pléiades triplet was acquired 1st May 2017, the day before the ASO flight and close to the accumulation peak" How different is that snowpack over a day, i.e., how much does the snow depth vary between scene acquisitions?

[Figure]

- Line 84 and after: I assume that the word "stable terrain" means a location that is snow free? This should be explicitly stated.

- Lines 26 versus 103 and 118: the abstract says "a snow-covered area of 137 km$^2$" while the Study site section says "a 280 km$^2$ subzone." Which area is it? Line 141: what about the "excluded 25 km$^2$" Line 306 states 138 km2

- Lines 124-125 versus Table 1: The text states a resolution of 2 m, while Table 1 says 0.5 m, which is it?

- The Methods are thorough. I suggest making sections 4.2 and 4.3 sub-sections of 4.1, as they are part of the overall Pléiades work-flow.

- The authors evaluate snow covered (Pléiades snow mask HS maps versus ASO HS greater than zero HS maps), and stable terrain (snow free) areas. What about the omission and commission areas? At least provide the percent of the study area for each of these.

- Lines 240-260: it is unclear why these equations are presented here. The paragraph begins with "For hydrological applications, HS maps are often spatially aggregated, for example to calculate the amount of snow in a catchment or an elevation band." Either change this sentence or add a sentence so we know what these equations are used for.

- Lines 263-265: it is not necessary to foreshadow what is in the Results "We first present the results for the HS maps calculated with the SGM-binary option and different image geometries. Then, we focus on the impact of the configuration of ASP. The best set of options and geometry is then used to analyze the spatial distribution of the residuals and to evaluate a model of the HS error." State clear objectives or research questions at the end of the Introduction and the reader would know what is to come.

- Tables 2 and 3: What is STD?

- Figure 4: I'm not sure if you mean corniche or cornice?

[Figure]

- Line 287: it should be "Artifacts." Also, the phrase: "Artifacts of typically 20 m x 20 m ..." is unclear

- Lines 306-307: what does "after erosion of the Pléiades snow mask" mean?

- Figure 9: it is unclear what the units in the y-axis are. The caption states: "h is the distance"

- This is also stylistic, but some sentence that tell what is upcoming and can be removed. For example, line 301-302 begins with "Figure 4 illustrates ..." The authors can just tell us the key point(s) in the Figure as the caption tells the reader what the figure is.

---

## Author Comment (AC1) · 30 May 2020

**Answer to Y. Bühler**

**The authors would like to thank Yves Bühler for his review. We provide below a point-by-point response to his comments and we explain how we intend to modify the manuscript in order to take them into account.**

Y.B . 1. Error combination of snow-on and snow-off DSMs: In their study they compare HS maps form Pléiades snow-on minus Pléiades snow-off with HS maps generated from LiDAR snow-on minus LiDAR snow-off. Certainly, this comparison makes sense and provides interesting insights but only as a second step. First, we need to see how much error is coming from individual surface models. If you take the snow-off and the snow-on DSM from the same platform and processing the errors of the individual DSMs could a) be added to each other or b) erase each other. So, if the snow-on DSM has a high error on one side (e.g. being too high) and the snow-off DSM has the error to the same side, the resulting snow depth is accurate even though both DSMs are bad. In the comparison as it is done now in the paper, there is now chance to see the quality of the individual DSMs. Therefore, I request the authors to do a HS comparison applying also the LiDARsnow-off DSM for the Pléiadessnow-on DSM. This will reveal how the snow-on DSM performs excluding random error addition from the snow-off DSM. Also, a DSM comparison between the Pléiades snow-off and the LiDAR snow-off DSM should be performed and discussed."

**We agree that an individual evaluation of each DSM can help understand the sources of error in the HS maps. We planned to compare Pléiades DEMs with ASO DSMs in order to identify how each dataset compare separately in snow-on and snow-off condition. The provider of the ASO data (JPL) could not provide us with the DSMs within the time of this review. Therefore we calculated a snow-on ASO DSM by adding the snow-off DTM and the ASO HS map. We used the Pléiades DEMs after the co-registration processing described in the present manuscript. We evaluated the areas labeled as snow at the snow-on date.**

**We include the figure S2, the table S3 in the supplement and this discussion in the 6.4 of the manuscript (L448):**

*"We further compared Pléiades snow-off DEM with the ASO snow-off DEM and Pléiades snow-on DEM with the ASO snow-on DEM. The latter was calculated by adding the ASO snow-off DEM and the ASO HS. Both Pléiades DEMs are co-registered as described in 4.1.4. We find a mean bias over snow-covered terrain of +0.13 m for snow-off conditions and +0.21 m for snow-on conditions (Table S3). These biases are of the same order of magnitude and suggest that a bias in the Pléiades snow-on DEM is partially compensated by the difference of the surface observed in the snow-off DEM (see above). In addition, the ASO snow-off DEM was acquired in October 2015 and the Pléiades snow-off DEM in August 2017. Growth or decay of the vegetation can occur over almost two years, leading to elevation differences between the snow-off DEMs. The NMAD is larger for snow-off DEMs (0.80 m) and snow-on DEMs (0.93 m) compared to HS residual (0.69 m). This shows that some errors are consistently present in the snow-off and snow-on DEMs of each type (airplane lidar or satellite photogrammetry). Pléiades DEMs are indeed over-estimating the surface elevation as the terrain slope increases (Figure S3). This suggests that combining satellite photogrammetry and airplane lidar DEMs may lead to larger errors than keeping homogeneous sourced DEMs."*

**Table S3.** Comparison of the snow depth residual (HS Pléiades minus HS ASO) and stable terrain elevation difference (Pléiades). All metrics are in meters. The bold line is common to this table and Table 2.

| | Area (km²) | Mean | Median | NMAD | RMSE | standard deviation |
|---|---|---|---|---|---|---|
| **difference of elevation difference (HS$_{Pléiades}$ minus HS$_{ASO}$)** | **138.02** | **+0.08** | **+0.10** | **0.69** | **0.80** | **0.79** |
| difference of elevation SNOW OFF (DEM$_{Pléiades}$ minus DEM$_{ASO}$) | 138.02 | +0.13 | +0.01 | 0.80 | 0.96 | 0.95 |
| difference of elevation SNOW ON (DEM$_{Pléiades}$ minus DEM$_{ASO}$) | 138.02 | +0.21 | +0.13 | 0.93 | 1.09 | 1.07 |

[Figure]

**Fig S2. Snow-off DEMs difference,Snow-on DEMs difference, HS difference (Pléiades minus ASO) against elevation (top), slope (middle) and aspect (bottom).**

2. Justification of selected photogrammetric processing method: The authors select three different processing methods in ASP (SGM-binary, SGM-ternary and Local search). But they do not justify why they selected these algorithms nor do they describe the algorithms and their differences. Are there other options? An overview on the available algorithms including their strength and weaknesses would be very helpful for the readers and would allow for the justification of the selection. Why does the SGM binary perform best? This section has to be expanded.

**Many more stereo algorithms and options are available in ASP to process stereo images. In fact we tested around 1500 combinations of six options in a small sub-zone of the study area in order to determine an optimal set of options. This analysis is computer-demanding and complex because all options are not compatible. In addition, the results for a small sub-zone may not apply to the entire study area. Since we could not find a clear message from this analysis, we decided not to publish it at this point. From these experiments we still learnt that the choice of the correlation algorithm and the cost function (binary census transform or ternary census transform) had a noticeable impact on the quality of the DEMs and HS maps. The comparison of these three sets of options already enables to highlight the impact of processing options on the HS maps. Future work will hopefully address more thoroughly this issue.**
**We added this L160:**
**"***These set of options were empirically selected but do not cover all the options available in ASP.***"**

**Explaining why one option performs better is beyond the scope of this study. However, we added some additional informations about the stereo algorithm and options L172:**
*"The SGM algorithm (Hirschmüller, 2005) differs from local-search window algorithm during the disparity map calculation. The local-search algorithm calculates the disparity for each pixel independently. The SGM algorithm optimizes the disparity over the whole image by assuming that disparity from neighboring pixels is likely to be close. This introduces more continuity in the disparity map and then in the DEM.*
*The matching of subsets of the images of a stereo-pair is measured with a cost function. The binary and ternary census transforms are two cost functions that convert a kernel centered on a pixel into a binary number. For the binary census transform, each pixel of the kernel is compared to the central pixel of the kernel and gives 1 if it is superior to it, 0 otherwise. All the digits are concatenated in a binary number associated with the central pixel. For the ternary census transform, each comparison of a pixel with the central pixel can give three different values: 00,01,11 depending on whether it is smaller, within, or greater than a buffer centered on the central pixel value."*

3. Comparison to results from other photogrammetric platforms (UAS and airplanes) and discussion of photogrammetry specific issues: The embedding of the work into the current literature is weak (section 6.1). It is true that they are among the first mapping snow depth with optical satellite data. But there has been a lot of additional work concerning photogrammetric snow depth mapping from UAS platforms (e.g. Van der Jagt et al. 2015, Bühler et al. 2016, De Michele et al. 2016, Harder et al. 2016, Adams et al. 2018). The authors should compare their results also to these publications and discuss more about the specific problems of photogrammetry on homogenous snow-covered areas (e.g. Bühler et al. 2017) and the effect of vegetation. I assume that there is distinct problem with noisy surfaces. This topic is not sufficiently discussed in the paper.

**We added references to UAVs work in the introduction (Bühler et al., 2016, De Michele et al., 2016, Harder et al., 2016, Redpath et al., 2018) and we added the paragraph below in the Discussion about the comparison between satellite photogrammetry and other airborne methods (airplane, UAV). However we feel that it is difficult to go beyond these generalities in our**

**manuscript. We prefer to refer to Eberhard et al. (2020) who made a detailed comparison of photogrammetric approaches using datasets from different instruments. See L506:**

*"HS maps from UAV SfM typically exhibit a centimetric bias (0.05 m to 0.11 m) and a RMSE between 0.05 m and 0.30 m based on comparison with snow probe and GNSS measurements. This is more accurate than what is currently achieved with satellite photogrammetry. However, UAV campaigns are currently limited to areas of less than 1 km² due to battery limitation and often relied on numerous ground control points. This greatly limits the possibility to cover large and remote areas. Airplane SfM exhibits accuracy close to UAV SfM with NMAD typically of 0.30 m (Bühler et al. 2015) and presents the same potential and logistic limitations as airplane laser scanning campaigns. The reader is referred to the study of Eberhard et al. (2020) for a detailed discussion on the different approaches to map snow depth with photogrammetry."*

**We also addressed the weakness of the embedding in the current literature by extending the section 6.1. We now include comparison of our results to McGrath et al. (2019), Shaw et al. (2019), Eberhard et al. (in TC discussion). These papers used satellite photogrammetry to measure at least the winter DEM. See L355 :**

*"**6.1 Comparison to others studies using satellite photogrammetry**

By comparing the Pléiades HS with the ASO data, we find a NMAD of 0.69 m in the best case (i.e. best acquisition geometry and ASP options), which is close to or higher than most previous evaluations (Table 4). Only Marti et al. (2016) measured a larger NMAD (0.78 m) with a reference HS map of 3.15 km² that was obtained by UAV photogrammetry. The spread in accuracy between studies in Tab. 4 could be due to differences in (i) the satellite data (i.e. acquisition geometry, image resolution), (ii) the characteristics of the study site and (iii) the representativeness of the validation data. The comparison with snow probes measurements showed NMAD about a third lower than this study at 0.45 m (n=442, Marti et al., 2016) and 0.47 m (n=36, Eberhard et al., 2020), but covered limited portions of the studied sites. The B/H for the images of Marti et al. (2016) range between 0.21 and 0.25 for all consecutive stereo pairs while our B/H range between 0.08 and 0.12. This is consistent with photogrammetry theory, which states that the accuracy of the DEM increases with the B/H up to a certain limit (Delvit and Michel, 2016). We find a similar NMAD to Eberhard et al. (2020) which calculated a HS map from a Pléiades snow-on DEM and an airplane SfM snow-off DEM and compared it to HS from airplane SfM only over 75 km² (NMAD=0.65 m). Finally, McGrath et al., (2019) found a NMAD of 0.24 m for HS from WorldView-3 stereo DEMs using 2107 point observations from ground penetrating radar surveys over a flat area of roughly 50 km². This lower NMAD value might result from the higher resolution of the WorldView-3 images (0.3 m). As the ASO provides a much larger reference dataset over a complex terrain, we argue that our study provides a more robust evaluation of the HS accuracy that can be expected from Pléiades in high mountain regions. While the ASO data itself may add some error, the published accuracy of the ASO HS data is significantly better than Pléiades. In all these studies, the absolute mean biases range between 0.01 m and 0.35 m."*

**Table 4.** *Comparison of HS accuracy with studies using satellite photogrammetry. *Eberhard et al. used a Pléiades DEM for snow-on and UAV or airplane SfM DEM for snow-off.*

| | Satellite (resolution) | HS map resolution | Validation data | Area | # | Mean | Median | NMAD | RMSE |
|---|---|---|---|---|---|---|---|---|---|
| This study | Pléiades (0.5 m) | 3 m | Airplane lidar | 138 km² | | 0.08 | 0.10 | 0.69 | 0.80 |
| Marti et al. 2016 | Pléiades (0.5 m) | 2 m | Snow probing | | 442 | | -0.16 | 0.45 | |
| | | | UAV SfM | 3.15 km² | | -0.06 | -0.14 | 0.78 | |
| McGrath et al. 2019 | WorldView (0.3 m) | 8 m | Ground Penetrating Radar | | 2107 | +0.01 | +0.03 | 0.24 | |
| Shaw et al. 2019 | Pléiades (0.5 m) | 4 m | Terrestrial lidar | 0.74 km² | | -0.10 | -0.22 | 0.36 | 0.52 |
| Eberhard et al.* | Pléiades (0.5 m) | 2 m | Snow probing | | 36 | -0.35 | -0.36 | 0.47 | 0.52 |
| | | | UAV SfM | 4 km² | | -0.18 | -0.18 | 0.38 | 0.44 |
| | | | Airplane SfM | 75 km² | | -0.02 | -0.18 | 0.65 | 0.92 |

YB. 4. Discussion on which applications can benefit from this approach and which not: With the elaborated accuracies in mind, there should be a discussion on which applications could benefit from this new method and which not. For example, regions with shallow snow cover (0 – 50 cm) over the vast regions on the higher latitudes of the northern hemisphere would be within the error range. So, I assume the method would be not accurate enough. But for what applications could it be of big value? I think this would be a very interesting part for the readers.

**As stated in the title of the article we think that this approach is mostly beneficial for mountainous terrain. However, we agree that we could further discuss its applicability in other contexts. We propose to include this paragraph (L515):**

*"6.7. Generalization to other regions*
*Several snow applications could benefit from HS maps from satellite photogrammetry. First this study could be reproduced in any place of the globe provided that i) there is a window to acquire snow-off images and ii) there is a way to co-register the series of DEMs, for example with stable terrain. This method is particularly suited for snow volume evaluation at a basin scale in alpine areas (this study site, Marti et al., 2016, McGrath et al. 2019, Shaw et al. 2019). Observing shallow snowpack (roughly HS below 50 cm, e.g. polar environments) might not be as straightforward as the typical spatial variability lays within our range of uncertainty (roughly 0.50 m). However, even landscapes with shallow snowpack often feature local accumulation of snow which would be measurable with satellite photogrammetry. Therefore it is hard to qualify this method as unfit to any region, but future studies are required to confirm its usefulness in these challenging contexts. Study of shallow snowpack would clearly benefit from higher accuracy DEMs through correction of the satellite jitter or increases in image resolution.*
*A lack of well distributed stable terrain in snow-on and snow-off DEMs can complicate the co-registration process in some regions. The horizontal component of the co-registration vector can be measured without differencing stable terrain and snow covered terrain (Marti et al., 2016) but the vertical component requires some stable terrain or an elevation reference. GCPs could be used but would limit the applicability of the method in remote mountains. Besides, it remains to be tested how many GCPs would be required and how precisely their position should be measured.*
*There are already a number of efficient free and open-access photogrammetric software tools that are under continuous development. These tools enable a high level of automation and are compatible with high performance computing environments (Howat et al., 2019). In our workflow, the last step to automate is the collection of training samples for image classification. This could be done by using an unsupervised classification algorithm or by using an external land cover classification. Preliminary work with a time series of Pléiades images in the Pyrénées (not shown here) suggests that it is not possible to simply use the classification model from a previous year to generate the classification of the current year. A possibility may be to use a Sentinel-2 snow cover map to extract training samples in the Pléiades multi-spectral images, since Sentinel-2 images have a shortwave infrared band which enables a robust and unsupervised detection of snow cover (Gascoin et al., 2019). Differentiating terrain covered with vegetation from stable terrain would remain challenging.*
*We find that the selection of the image configuration and the processing options can lead to changes in the NMAD up to ~0.3 m. Fig. 10 suggests that this variation is likely to become insignificant if the HS map is aggregated at a larger spatial scale (grids spacing larger than 100 m x 100 m). Such optimisation is therefore more important for the study of small-scale features (wind drift, avalanches, typically about a few tens of meters) or to decrease bias on specific terrain (south slopes, fields with isolated trees). The optimization of the photogrammetric processing can also be important when little stable terrain is available for the co-registration step."*

Detailed comments:

P3L65: Why are DSMs less accurate in steep terrain? Is it mainly because a small x,y shift results in a large z-shift? Please explain a bit more

**DSM are expected to be less accurate in steep terrain because a small error in horizontal positioning results in a large vertical error. Horizontal errors are inherent to the photogrammetric process (see below the answer to comment about P13L325). Steep slopes might also be more prone to error as the image distortion also depends on the local incidence angle. Two images will look more similar on areas taken with small incidence angles than large incidence angles. Image matching process should be more accurate when images are more similar.**

**We modified the text for (L66):**

*"This lack of validation data in steep slope areas was an important limitation of this study since DEMs from stereoscopic images are known to be less accurate on steep slopes due to a higher sensitivity to horizontal error and to local image distortion (Lacroix, 2016; Shean et al., 2016)."*

P3L76: the term airborne lidar altimetry sound a bit strange to me, I would use airborne laser scanning ALS.

**We agree with the reviewer and replaced references to airborne laser scanning with ALS.**

P3L87: Please give some references for this statement

**We deeply modified this paragraph as we now evaluate a similar but well-accepted error model (Rolstad et al., 2009) L243:**

*"The accuracy of HS maps is often discussed at (or close to) the highest resolution that is allowed by the sensor (e.g. Nolan et al. 2015, Marti et al., 2016). In practice however, HS maps may be subject to spatial averaging to assimilate in a snowpack model, to estimate catchment-scale HS or to compare with coarser satellite products and model output (Painter et al., 2016; Margulis et al., 2019; Shaw et al., 2019). The accuracy of the mean HS of a set of contiguous pixels is expected to be higher than a single pixel accuracy but depends on the spatial correlation of the errors (Rolstad et al., 2009)."*

P4L103: Why did you select exactly this zone? Can you give some justification?

**We selected this zone to ensure a large overlap between Pléiades and ASO coverage while including a large range of elevation. We modified the text (L92):**

*"The ASO flights cover 1100 km² in the basin while this study focuses on a 280 km² subzone which was selected to cover a large elevation range."*

P4L118: How was the image acquisition performed? What prices did you have to pay? What would be the conditions for people who want to do the same as you did?

**We would now include this paragraph L117:**

*"Pléiades images were obtained at no cost through the DINAMIS program (https://dinamis.teledetection.fr/) opened to Europeans scientists working in public research institutions. Otherwise Pléiades images can be ordered from Airbus Defense and Space."*

**We prefer not to indicate the commercial cost of Pléiades imagery because it may be subject to change. The interested user can easily contact ADS to request a quote.**

P5L135: Why is the resolution 3 m is there a justification for that? What was the point density per m2 of the LiDAR dataset? Double period at the end of the sentence.

**We used ASO products described in Painter et al. (2016) and could not chose the HS map resolution. The 3m resolution balances computational efficiency and the lidar point density,**

**which varies from just under 1 pt/m2 in the worst case to > 10pts/m2 at highest terrain elevations, due to the constant flight altitude of the ASO aircraft. We added the reference to Painter et al. (2015) to make clear that we did not process these data L127:**
*"Ground points are aggregated to a 3 m grid to derive a gridded DEM (Painter et al., 2015)."*

P7L173: In my experience it can be very hard to find snow free, stable terrain in high alpine regions and this with a good distribution over the entire investigation area. Please discuss this issue and give some recommendations what to do if not enough stable terrain can be found. Please also discuss why an absolute orientation via GCPs is not suitable.
**We agree that stable terrain availability can be a limitation to this method. We identified areas with bare rock or grass as stable terrain because it was available in this study area. However this type of terrain is not necessarily always available in both snow-on and snow-off images. As in Marti et al. (2016), the horizontal component of the co-registration vector can be measured without differencing the stable or snow covered terrain. The vertical component of the co-registration vector should ideally be measured on well-distributed stable terrain (this study), but if this is not available, can be performed using stable terrain in a specific location of the images (e.g. a football field in Marti et al., 2016, snow-free road). If no stable terrain is available one could think of using snow depth measured in the field (snow probe, weather station).**
**The use of accurate and well-distributed GCPs improves the accuracy of individual DEMs (Berthier et al., 2014). However, we did not explore how many GCPs and what precision would be necessary to measure snow depth. The clear advantage of a method without GCP is that it can be readily applied elsewhere on Earth.**
**We added this paragraph in the discussion to comment on that point (L527):**
*"A lack of well distributed stable terrain in snow-on and snow-off DEMs can complicate the co-registration process in some regions. The horizontal component of the co-registration vector can be measured without differencing stable terrain and snow covered terrain (Marti et al., 2016) but the vertical component requires some stable terrain or an elevation reference. GCPs could be used but would limit the applicability of the method in remote mountains. Besides, it remains to be tested how many GCPs would be required and how precisely their position should be measured."*

P7L176: Why did you choose the thresholds -1 to 30 m? Please justify
**These thresholds are used to remove obvious outliers in the DEM difference map. Minus one (-1 m) is set to take into account that Pléiades elevation difference might be negative despite presence of snow due to its accuracy. Plus thirty (+30 m) is a conservative estimate based on expert judgment of the snow depth upper bound. We added this L207:**
*"Pléiades HS values below -1 m and above 30 m were set to no data to exclude unrealistic outliers based on expert judgment and considering the minimal value that Pléiades HS could reach for actual HS close to zero."*

P7L182: In my experience a big problem are alpine bushes (0.5 to 2 m high). How do you treat these? It is not mentioned
**We did not apply specific treatment for bushes, their impact is included in our residual calculation. We expanded comments on the impact of vegetation on our evaluation L425:**
*"We found a mean difference of +0.08 m between Pléiades (SGM-binary, front-nadir-back) and ASO HS despite the correction of the vertical offset between the snow-on and snow-off DEM using stable terrain after co-registration. This bias is low given the differences in the characteristics of the ASO and the Pléiades products. It can be due to many factors including the effect of vegetation. First, the ASO snow-off DEM is a digital terrain model while the Pléiades snow-off DEM is a digital surface model. Tall vegetation (i.e. trees) is identified during the classification of the MS images and do not impact the HS evaluation. But short vegetation completely covered with snow in winter is not identified in the classification. For ASO products, filtering based on the multiple lidar returns*

*produced by vegetation should provide the ground elevation, but short vegetation often does not produce multiple returns (Painter et al., 2015). Furthermore, there is a large known error in vegetation height measured with Pléiades DEMs (Piermattei et al., 2018). Thus, it is still unclear which surface is sensed by each method between the top of the vegetation and the underlying ground."*

P7L190: Why do you use nearest neighbor and not cubic resampling? Please justify.
**We used nearest neighbour interpolation for the masks as they are binary, each pixel being one or zero. Other interpolation schemes (cubic, linear) would require another step of thresholding to obtain again a binary mask.**

P10L244: Artifacts in the equations, please check all equations
**The equations were replaced and should now be readable.**

P11L266: It would be very helpful for the readers to have a figure showing the matching success and the hillshade of the DSMs generated with the different pairs This would also clarify the artifacts you mention.
**The figure below shows the hillshade obtained with the different processing options, highlighting the described artifacts. We will add it in as a supplement to a revised version of this manuscript.**
**We unfortunately did not keep intermediate products of ASP, including the matching success maps. However, from our experience hillshade maps are more informative than matching success maps to visualize the impact of ASP configuration like these artefacts. We are not sure if the added value of a figure showing the matching success justifies the computational cost of running ASP again but we are ready to do it if the referee thinks it is important. We also added a description of the artifact L289:**
*"Patches of typically 20 m x 20 m with abnormally large HS (>10 m) compared to ASO (~3 m) are also observed with the Local-Search options around isolated trees. These artifacts are not visible with the SGM-binary or ternary options (Figure S1)."*

[Figure]

**Figure S1**. Hillshades of the snow-on Pléiades DEM calculated with the stereo options Local-Search (a), SGM-binary (b) and SGM-ternary (c). Artifacts around isolated tree were observed with Local-Search (arrow in hillshade (d) and pan-chromatic ortho-image (g)). Geometrical artifacts were observed in south facing slopes with SGM-ternary (arrow in hillshade (f)).

P12L296: Why do you use the threshold 3 x NMAD? Please justify.

**This filtering is used to remove wrongly labeled pixels only in the co-registration process. It can for example happen that forest were included in stable terrain. If the residuals were following a Gaussian distribution, points further away than 3 x NMAD should occur less than 1% of the time. Hence such points are considered as very unlikely and excluded as outliers. This is an adaptation of the 3 x standard deviation filtering (Nuth and Kääb, 2011) with a metric adapted to elevation difference residual (NMAD, Höhle and Höhle, 2009). This type of filter is also sometimes used before averaging elevation aggregated by elevation but must be carefully used as it can eliminate valid extreme value and thus introduce bias (Dussaillant et al., 2019).**
**This is stated L298:**
*"This is expected as the same filtering is used during the co-registration process to remove outliers."*

P13L325: Do you have a hypothes is where this error is coming from?

**Random error could result from several phenomena. For example the fact that the images are a discrete representation of a continuous surface. This means that the images, the DEMs or intermediate products are necessarily interpolated during some operations. It is the case during the sub-pixel refinement in the images correlation. Another source of error is that the images are not corrected for the atmospheric or BRDF effect on the radiometry.**

P13Table2: There is a 10 cm difference in RMSE and NMAD between front-nadir-back and nadir-front-back. Why is this showing, please explain as you take basically the same input data just in reversed order.

**The processing in ASP depends on the order of the images. For a triplet of images (A-B-C), ASP calculates image matching for two pairs of images (A-B and A-C) defined by the order of the images. As the B/H ratio is different for each pair, the DEMs will be different depending on the order.**
**We expanded these explanation L188:**
*"In the tri-stereo case, ASP calculates two disparity maps and performs a joint triangulation when calculating the point-cloud. In the first tri-stereo case (front-nadir-back), ASP calculates a disparity map between the front and the nadir image and between the front and the back image. In the second case (nadir-front-back), ASP calculates a disparity map between the nadir and the front image and between the nadir and the back image. The order of the images matters in the tri-stereo case since the B/H is different between front-nadir and front-back, or nadir-back and front-back. We did not evaluate the third possible tri-stereo combination (back-nadir-front) as we expect results to be similar to the front-nadir-back case."*

P14L346: Please extend this section substantially by discussing results from previous UAS and airborne investigations including the reached accuracies and performance on snow covered surfaces (see my main point 3)
**We expanded this section as described in our answer to point 3.**

P14L362: But please mention the benefit of additional coverage in particular within steep slopes.
**We agree and added this L383:**
*"Tri-stereo might provide greater benefits in case of image occlusion in steep slopes, which is more prone to occur with higher B/H."*

P14L366: Can you make the statement "B/H around 0.2 seems to be beneficial" stronger by providing more justification
**We modified the text in L387:**

*"From these two studies and for similar terrain, a triplet of images with a B/H for consecutive images around 0.2 seems a good compromise. It should ensure high coverage and good DEM precision. Further work is needed to confirm this statement, by testing varying B/H values."*

P15L400: what was calculated ASO minus Pléiades or the other way round? ASO should be higher as it uses a DTM as snow-free DSM

**This is calculated as Pléiades minus ASO. As suggested ASO HS should be higher as it uses a DTM. This implies that there is a larger positive bias on Pléiades HS than what we measure with the HS comparison. This is confirmed by the comparison of snow-on DEMs which shows that Pléiades snow-on is on average 0.25 m above ASO snow-on over snow terrain (see above answer to main comment 1.).**

P17L449: Please expand this section taking into account published UAS and airborne photogrammetric investigations

**We expanded this section as described in our answer to point 3.**

P18L473: The conclusions are a bit thin. Please also provide the RMES values. Could you provide more important information? The term "good accuracy" is vague.

**We expanded the conclusion as follow L551:**

*"7 Conclusion*

*We found a good agreement between snow depth (HS) maps from high resolution stereo satellite images with ALS HS maps over 138 km² of mountainous terrain in California. The mean residual is +0.08 m, the NMAD is 0.69 m and the RMSE is 0.80 m. Comparison of individual DEMs show a growing positive bias with slope in Pléiades DEMs. This bias is of similar magnitude in both snow-on and snow-off Pléiades DEMs and thus cancel out in the HS map, leading to agreement between Pléiades and ALS HS for all slopes up to 60°. South facing slopes seem prone to a positive bias in the Pléiades HS (~0.2 m). These areas were found to have less texture in the panchromatic images. The main drawbacks of the satellite stereo HS method are the lack of data under dense tree cover, the reduced accuracy in shaded areas, and the current challenge to image large regions in a short time. We found that the accuracy of the maps was sensitive to the B/H and the photogrammetric processing options. Using the current ASP multi-view triangulation routines, we could not find a clear benefit from the use of a triplet of images compared to a pair with optimal B/H (about 0.2). The accuracy of the HS maps can be improved by decreasing the resolution. This improvement cannot be described with a well-accepted statistical model partly due to an undulation pattern commonly observed in DEMs derived from satellite photogrammetry. We observe that the accuracy is improved by 50 % when decreasing the HS maps resolution from 3 m to 36 m. We conclude that satellite photogrammetric measurements of HS are relevant for snow studies as they offer accuracy of ~0.70 m at 3 m resolution, a high level of automation and the potential to cover remote regions around the world."*

P20Fig4: The scale bar is too large hindering a sound interpretation of the values. I would propose a range of 0 to >5 m. Here a difference image (HS_ASO minus HS_PLéiades would be essential. This would allow for a detailed interpretation of the results, also for the subsets. What you name an avalanche deposit in d) does not look like that to me at all. I saw many avalanche deposits in HS maps and I am pretty sure that this one is something else. Please check that. Why did you locate the transect there? Can you give a justification?

We provide below the Pléiades snow height map with a color bar ranging from 0 to 5 m. More areas appear saturated. We are not sure whether this makes the map more readable but would follow the editor's opinion.

The difference image is already provided in Fig. 8. It seems to us that figure 4 would get harder to read if a panel was added. We now refer to it in the caption of Fig. 4 so that reader can see the link.

We provide a zoom on what we called an avalanche in Fig. 3. We made this supposition based on the interpretation of the topography (steep couloir and flat area at the bottom), the pan-chromatic image (texture indicating dynamic deposition) and the snow height (accumulation at the bottom of the couloir and upward). We are still of the opinion that it is an avalanche. We agree that the arrow we positioned on the figure was misleading as it points at the couloir. We now moved it so that it points at the deposition area. Similar observation that we also interpreted as avalanche deposition are visible below in profile B.

[Figure]

Fig 3. Zoom on the avalanche deposit in the pan-chromatic image (left) and snow depth map (right).

[Figure]

**Fig. 4 with the colorbar updated to the range [0,5 m].**

P21Fig5: Here it would help if you could draw some more transects at further locations to see the differences

**We drew more profiles trying to cover different ranges of snow depth and vegetation condition. HS seems more noisy and less accurate in forest environment (profile A, D, E). Another case of what seems to be avalanches deposit is visible in profile B.**

**We propose to add these profiles in the supplement. We mention it in Fig. 5 caption L601:**

*"**Figure 5.** Transect T-T' of snow depth visible on Figure 4. e. from Pléiades data (pink) and ASO (blue). More transects are available in supplement."*

[Figure]

[Figure]

[Figure]

[Figure]

[Figure]

[Figure]

P21Fig6: The ASO HS median is mostly lower than the Pléiade HS even though the ASO uses a DTM as snow-free surface (penetrating low vegetation). Therefore, we would expect the LiDAR HS to be slightly higher. Can you comment on that?

**We agree with this suggestion. We did not take into account the fact that ASO snow-off elevation is a DTM while Pléiades snow-off elevation is a DSM although we do not know precisely which surface is represented by Pléiades elevation. Evaluation of the residual against vegetation height suggest that Pléiades do not represent the top of the vegetation at least for low vegetation. Further work should evaluate this point.**

**Following the suggestion of comparing individual DEMs (snow-on and snow-off) separately we conclude that Pléiades snow-on DEM bias of +0.21 m is partially compensated by the fact that we used a DSM for snow-off, resulting in a lower bias on HS +0.08 m. This is commented in answer to the point 1.**

**We also added this in the discussion L425:**

*"We found a mean difference of +0.08 m between Pléiades (SGM-binary, front-nadir-back) and ASO HS despite the correction of the vertical offset between the snow-on and snow-off DEM using stable terrain after co-registration. This bias is low given the differences in the characteristics of the ASO and the Pléiades products. It can be due to many factors including the effect of vegetation. First, the ASO snow-off DEM is a digital terrain model while the Pléiades snow-off DEM is a digital surface model. Tall vegetation (i.e. trees) is identified during the classification of the MS images and do not impact the HS evaluation. But short vegetation completely covered with snow in winter is not identified in the classification. For ASO products, filtering based on the multiple lidar returns produced by vegetation should provide the ground elevation, but short vegetation often does not produce multiple returns (Painter et al., 2015). Furthermore, there is a large known error in vegetation height measured with Pléiades DEMs (Piermattei et al., 2018). Thus, it is still unclear which surface is sensed by each method between the top of the vegetation and the underlying ground."*

---

## Author Comment (AC2) · 30 May 2020

**Answer to P. Harder**

The authors would like to thank Phillip Harder for his review. We provide below a point-by-point response to his comments and we explain how we intend to modify the manuscript in order to take them into account.

P.H. This work provides a deeper evaluation of the snow depth mapping from stereo satellite imagery approach proposed by Marti et al (2016). The advancement of this work is to consider some of the various/emerging DSM processing options and a more thorough error analysis with basin scale airborne lidar data from ASO versus manual discrete manual probe observations in locations not fully representing the variability present in the landscape. The prospect of obtaining high resolution (3m scale here) snow depth with errors reported here to be less than 0.8m from a space borne platform is tremendously exciting and the benefits of such a capability are clearly articulated herein better than I can summarise in this space.

Much of this article is clear and well written but there are a couple aspects which would benefit from some clarification and/or clearer justifications. I will begin with some main comments and then provide a list of more technical comments/edits/suggestions. Overall I think this work is well suited to The Cryosphere. The previous review of Buhler makes many important observations which I fully agree with. I would highly recommend the authors make those edits in addition to some more articulated here.

Main Comments:

1) Error model: there is a lot of discussion of methodology and results of the scaling of the random error with length scale. I have a couple concerns on this. First we are looking at the random error metric, articulated later on as the standard deviation of the snow depth residual error. This is only one part of the error, as in overall error is comprised of random components (captured here) as well as biases (not a part of this model). Correct me if I am wrong but my read is that increasing length scales will lead to decreases in random error and will therefore not comment on the bias error? Or, this model shows that increasing the length scale will increase precision but does not say anything about the accuracy? We could have really large biases in the dDEM but these will not be reflected in the model? A quick read from the abstract doesn't articulate this nuance that I am perceiving. Without lidar or ground observations to correct for the bias this suggests there will be operational challenges to implement this method in data sparse regions. Would co-registration on common snow-free stable areas be a reasonable approach to provide these relative differences? Would you have any other suggestions to improve applicability in data sparse areas?

We modified the error model to use the one proposed in Rolstad et al. (2009) which is a more rigorous and better defined than what was shown in the initial manuscript. We deeply modified the presentation of it (Method 4.2, Results 5.5, Discussions 6.5) and hope that it is more understandable now.

Anyway this model indeed concerns random spatially correlated error. The version we use assumes that the data are not biased. We first compare it to uncorrected HS residual to match what one would obtain on another study site without validation data and without the possibility to unbias or detrend the data. We then compare it to corrected HS to measure the impact of the satellite jitter. We now comment on what level of bias should always be expected with satellite photogrammetry (L487):

*"Finally, although the bias or systematic error is corrected on stable terrain, there remains a bias on HS of the order of ~0.20 m (Table 4) that should be taken into account in the error calculation. According to the literature, this bias can be estimated by comparing the mean and median of elevation differences over stable terrain (Gardelle et al., 2013) or by calculating the residual of co-registration vector when more than two elevation datasets are available (Nuth and Kääb, 2011)."*

**About co-registration in data-sparse areas, we added a paragraph in discussion L527:**
*"A lack of well distributed stable terrain in snow-on and snow-off DEMs can complicate the co-registration process in some regions. The horizontal component of the co-registration vector can be measured without differencing stable terrain and snow covered terrain (Marti et al., 2016) but the vertical component requires some stable terrain or an elevation reference. GCPs could be used but would limit the applicability of the method in remote mountains. Besides, it remains to be tested how many GCPs would be required and how precisely their position should be measured."*

Second, there is an assumption that stable terrain residual random errors apply to snow surfaces. Is this a valid assumption? In most topography the snow surface will reflect the underlying ground surface. In complex alpine topography the range in surface elevations can be orders of magnitude greater than the hs corresponding to the dDEMs and therefore this is a reasonable assumption. But in many areas prone to wind redistribution parts of the landscape can be smoothed out, for example, wind blown snow fills up gullies.
Therefore, I would expect the random components of the error to vary between snow and snow-free area. While the ability to predict error of a snow depth product is of interest I worry that this is perhaps a little too simplistic and detracts from the main point of the paper which is a detailed evaluation of satellite stereo imagery snow depth measurement and its comparison to airborne lidar data. The comments from Buhler about partitioning the error the errors of the snow-free and snow covered surfaces would be very valuable and provide a clearer interpretation of what the HS error is derived from.

**We agree that many reasons point to the fact that stable terrain residual and HS residual might not be equal. However there is often no other choice than using statistics over stable terrain to estimate uncertainty of HS. We try to evaluate this approach in 6.5 comparing either statistics of stable terrain or HS in the error model with the measured residual at different resolutions. We comment on that in discussion L481:**
*"This analysis shows that the model proposed by Rolstad et al. (2009) provides a good first order estimation of the random error after spatial aggregation under the assumption that there is no spatial drift in the error at scales beyond the correlation length. In most cases, the statistics of the HS residuals are not available and might be only measured on stable terrain. Interestingly in this study, the length of correlation of the error is similar over stable terrain and snow terrain. However, the dispersion (NMAD, standard deviation) is two folds larger over snow covered terrain than stable terrain, which leads to a proportional underestimation of the error."*

**We now comment on individual DEMs evaluation. We include figure S2 and table S1 in the supplement and this discussion in section 6.4 of the manuscript L448:**
*"We further compared Pléiades snow-off DEM with the ASO snow-off DEM and Pléiades snow-on DEM with the ASO snow-on DEM. The latter was calculated by adding the ASO snow-off DEM and the ASO HS. Both Pléiades DEMs are co-registered as described in 4.1.4. We find a mean bias over snow-covered terrain of +0.13 m for snow-off conditions and +0.21 m for snow-on conditions (Table S3). These biases are of the same order of magnitude and suggest that a bias in the Pléiades snow-on DEM is partially compensated by the difference of the surface observed in the snow-off DEM (see above). In addition, the ASO snow-off DEM was acquired in October 2015 and the Pléiades snow-off DEM in August 2017. Growth or decay of the vegetation can occur over almost two years, leading to elevation differences between the snow-off DEMs. The NMAD is larger for snow-off DEMs (0.80 m) and snow-on DEMs (0.93 m) compared to HS residual (0.69 m). This shows that some errors are consistently present in the snow-off and snow-on DEMs of each type (airplane lidar or satellite photogrammetry). Pléiades DEMs are indeed over-estimating the surface elevation as the terrain slope increases (Figure S3). This suggests that combining satellite photogrammetry and airplane lidar DEMs may lead to larger errors than keeping homogeneous sourced DEMs."*

**Table S3.** Comparison of the snow depth residual (HS Pléiades minus HS ASO) and stable terrain elevation difference (Pléiades). All metrics are in meters. The bold line is common to this table and Table 2.

| | Area (km²) | Mean | Median | NMAD | RMSE | standard deviation |
|---|---|---|---|---|---|---|
| **difference of elevation difference (HS$_{Pléiades}$ minus HS$_{ASO}$)** | **138.02** | **+0.08** | **+0.10** | **0.69** | **0.80** | **0.79** |
| difference of elevation SNOW OFF (DEM$_{Pléiades}$ minus DEM$_{ASO}$) | 138.02 | +0.13 | +0.01 | 0.80 | 0.96 | 0.95 |
| difference of elevation SNOW ON (DEM$_{Pléiades}$ minus DEM$_{ASO}$) | 138.02 | +0.21 | +0.13 | 0.93 | 1.09 | 1.07 |

2) Structure: There is a lot of detailed technical discussions but many sections of this paper would benefit from stepping back for a moment and explaining the justification for what is occurring and how it fits in to the overall story. As it is there are some disjointed sections. (2 examples: line 125-126 – this sentence requires context, line 240 onwards – why do we care about developing an error model?)

**We modified many parts of the article to make the scientific goals clearer for instance in the introduction (L77). We also modified the error model calculation and presentation. Its use is now justified in introduction (L84). We think that presentation of the error model makes sense in this article as it makes use of this rare datasets combination and is of interest for future similar studies. We present the error model L243 and associated results L331 and L463.**

Specific comments:
Line 21 "A recent new method" -> "Recently, a method"
**We modified according to the reviewer's suggestion (L21).**

Line 32: "up to" -> "down to"? I don't see anywhere else in the paper where it is mentioned that there is a factor of two decrease in random error going from 3m to 20m (short of a reader interpretation of Figure 10). Also as a sample size increases the standard deviation (aka random error here) will always decrease – can you articulate the nuance that the modelling modifies this relationship by accounting for spatial correlation?

**It is true that we did mention this interpretation of Figure 10 out of the abstract. We added it in the model error results L332:**

*"The measured error of the HS map decreases with increasing resampling resolution (Fig. 10). The NMAD of the HS residuals is reduced by a factor of almost two by resampling from the original resolution of 3 m (NMAD=0.69 m) to 36 m (NMAD=0.38 m)."*

**We agree that the random error decreases as the sample size increases. The point of the error model is to define at which rate it decreases with the sample size. This especially depends on the length of correlation of the error. We tried to explain this and why we use Rolstad et al. (2009) error model in the rewritten method (L243):**

*"The accuracy of HS maps is often discussed at (or close to) the highest resolution that is allowed by the sensor (e.g. Nolan et al. 2015, Marti et al., 2016). In practice however, HS maps may be subject to spatial averaging to assimilate in a snowpack model, to estimate catchment-scale HS or to compare with coarser satellite products and model output (Painter et al., 2016; Margulis et al., 2019; Shaw et*

*al., 2019). The accuracy of the mean HS of a set of contiguous pixels is expected to be higher than a single pixel accuracy but depends on the spatial correlation of the errors (Rolstad et al., 2009)."*

Line 35-37: A great conclusion!

Line 44: Nolan et al. 2015 is not a UAV reference, they use SfM from a manned airplane. Other early reference options would be Buhler et al. 2016, Harder et al. 2016 or De Michele et al. 2016 for starters.
**We agree and replaced the Nolan et al. 2015 reference with the ones suggested by the reviewer (L41).**

Line53-55: One criticism from abstract was that the Marti paper only considered limited numbers of observations and terrain representation with manual insitu probe depths while here there is acknowledgment that there was also validation versus UAV data? Clarify this contradiction?
**We agree that this contradiction is misleading. We modified the abstract (L23) in:**
*"However, the validation was limited to probe measurements and UAV photogrammetry, which sampled a limited fraction of the topographic and snow depth variability."*

Line 72: Harder et al. 2016 also showed that SGM improved performance over low texture snow in the UAV-SfM snow depth mapping context.
**We missed this information. We now added the reference to Harder et al. 2016 (L75).**

Line 136-137: this is not a method to do forest snow depth mapping so is it important to include forest lidar data processing steps?
**We agree with the reviewer and removed this sentence and the following one explaining how SWE is calculated in ASO program.**

Line 162: if including coordinate system information should also put in the projection.
**We now state clearly that we used datum WGS84 L153:**
*"The output DEM resolution and coordinate system was defined to match those of the ASO product (UTM 11 north, WGS 84)."*

Line 175: What was the extent of the Pleiades HS values that was outside these thresholds?
**We mention this result in results section L306:**
*"No HS were higher than 30 m but 0.25 km² of HS were excluded because HS was less than -1 m. This occured in areas covered with low density deciduous vegetation which was classified as snow."*

Line 188: Is "eroded" the proper term to describe this? I feel this may be confusing for those with geomorphology mindsets.
**Erosion is a term used in image morphology which can be indeed confusing. We replaced this sentence with L221:**
*"The stable terrain and snow masks were shrunk (morphological erosion) with a radius of two pixels (4 m) and patches smaller than 30 pixels (270 m²) were removed."*

Figure 2: Line passes through the "co-registration" step. Does this need a box or to be offset like other processing step descriptions?
**We modified the figure so that no line crosses the processing step description.**

Line 240 -255: empty super and sub script boxes appear in many of the equation/symbol text. Add space between equations and the equation numbers as well?
**The equations were replaced and should appear normally now.**

Section 5.1. Can you clarify the results and discussion around the comparison of pairs versus triplets? In parts (like the last sentence of this section you say the triplet is the best) while in others (line 360) it is justified that a pair of images is just fine. Line 382-384 says tri-stereo is best to provide the best coverage and reduce distortion. Can this be clarified to allow for more consistency throughout?

**We agree that we did not handle conclusion about the benefit of bi- or tri- stereo clearly. We do not find a clear benefit of using tri-stereo images in this study case since front-back images resulted in snow depth with comparable accuracy and no data gaps. However, we do not think that this will hold for any terrain, image acquisition angle (B/H of the front-back pair) and processing workflow.**

**We tried to make this clearer through the manuscript.**

**For instance, L281:**

*"In the following sections, the HS map from the front-nadir-back geometry is used as it yielded the lowest bias, RMSE and NMAD of all the geometries although similar to the front-back geometry."*

**and L382:**

*"We do not find a large added-value of the tri-stereo images for the map accuracy compared to an optimal bi-stereo configuration. Tri-stereo might provide greater benefits in case of image occlusion in steep slopes, which is more prone to occur with higher B/H."*

Line 320-325: the low frequency undulation in HS residuals. Where is this coming from? How pervasive is this error with satellite stereoscopy or is it specific to this site/processing options? For this approach to be of value what techniques can be employed to address this error (while low amplitude could be important) where there is no ASO like lidar data for validation?

**These error are likely due to satellite movement (jitter) not measured by the onboard device and thus unmodeled in the RPCs. We describe this L326:**

*"Such undulation pattern was observed in other Pléiades products, ASTER images (Girod et al., 2017) and World-View DEMs (Fig. 10 in Shean et al., 2016, Fig. 6 in Bessette-Kirton et al., 2018). It is attributed to unmodeled satellite attitude oscillations along-track (jitter)."*

**This kind of error can occur in any site and at least with optical satellites which acquire images with a pushbroom sensor. This effect can be mitigated by identifying the jitter in the elevation difference map either on stable terrain (Girod et al., 2017) or over the complete area in rare cases (this study). ASP has a module to handle the jitter effect at the triangulation step which calculates the point cloud from the disparity map. However this module is "highly experimental" according to ASP documentation (https://stereopipeline.readthedocs.io/en/latest/index.html on april 29th). Therefore we did not make any attempt to use it. We added a reference to Girod et al. 2017 (L467):**

*"To verify this explanation, we applied an empirical correction to remove the undulation pattern from the residuals map. We averaged the HS residuals by pixel rows in the across-track direction and used a Fourier transform to identify the undulation frequencies (adapted from Girod et al., 2017). Then, we modelled this error by selecting the frequencies lower than $4 \; 10^{-4} \; m^{-1}$ (i.e. wavelength longer than 2.5 km) and removed it from the HS map."*

Line 330-334: are you applying the error model to the undulation removed HS residual or the raw HS residual? Can you clarify that? If you consider a semivariogram that extends out to the amplitude of the undulation does the undulation length scale appear in the semivariogram (extending figure 9)

**Figure 9.b. now shows the semi-variogram for a larger range of distance. The impact of the undulation and its correction is visible and commented L322:**

*"The semi-variogram of the residual increases from 0.2 to 0.8 linearly for lag distances between 3 and 20 m (Fig. 9.a). Low amplitude undulation for lag distances between 2000 m to 8000 m (Fig. 9.b) are related to a low frequency undulation in the HS residual map, which has an amplitude of*

*approximately 30 cm and a wavelength of about 4 km (Fig. 8). The crests of the undulation are oriented in the east-west direction (Fig. 8). Such undulation pattern was observed in other Pléiades products, ASTER images (Girod et al., 2017) and World-View DEMs (Fig. 10 in Shean et al., 2016, Fig. 6 in Bessette-Kirton et al., 2018). It is attributed to unmodeled satellite attitude oscillations along-track (jitter). A similar semi-variogram shape is obtained over stable terrain. From this semi-variogram analysis we estimate that the correlation length of the residuals (see 4.2) is about 20 m for both snow and stable areas."*

**In part 5.5 we apply the error model to the raw HS residual (not corrected). We tried to make it clearer L332:**
*"The measured error of the HS map decreases with increasing resampling resolution (Fig. 10). The NMAD of the HS residuals is reduced by a factor of almost two by resampling from the original resolution of 3 m (NMAD=0.69 m) to 36 m (NMAD=0.38 m). As explained in Sect. 4.2, we computed two error models using either the HS residuals (= 20 m, = 0.69 m) or the stable terrain residuals (= 20 m, = 0.40 m) to parameterize Eq. (1) and (2). We find that the NMAD of the HS residuals matches well the error modelled in for averaging areas smaller than $10^3$ m² when (, ) are calculated with the HS residuals (Fig. 10). However it does not match with the modeled error for averaging areas larger than $10^3$ m² (Fig. 10). This is due to the lower decrease of the residuals dispersion with spatial resolution. The measured NMAD decreased by 0.07 m between 36 m resolution and 180 m resolution while the modeled error decreased by 0.22 m between the same resolutions. We attribute this mismatch to the undulation pattern identified in Sect. 5.3 (see Sect. 6.5 in Discussion)."*

**We apply the error model to the corrected HS residual in Discussion, part 6.5, L464.**
*"The error predicted with Eq. (1) and (2) does not agree with the NMAD of measured HS error for averaging areas larger than $10^3$ m² (Fig. 10). This is likely because Eq. (1) assumes a randomly distributed error beyond the short distance correlation length (here 20 m), while the undulation pattern identified in Fig. 8 introduces an additional spatial correlation at larger scales in the HS residuals map. To verify this explanation, we applied an empirical correction to remove the undulation pattern from the residuals map. We averaged the HS residuals by pixel rows in the across-track direction and used a Fourier transform to identify the undulation frequencies (adapted from Girod et al., 2017). Then, we modelled this error by selecting the frequencies lower than $4\ 10^{-4}$ $m^{-1}$ (i.e. wavelength longer than 2.5 km) and removed it from the HS map. As expected, this correction makes the semi-variogram of the HS residual flatter for lag distances between 2000 m and 8000 m (Fig. 9.b). As a result, there is a better agreement between the HS residuals NMAD and the modeled error with and $l_{cor}$ estimated from the HS residuals ($l_{cor}$ = 20 m, = 0.69 m) (Fig. 10). The improvement is more marked at lower resampling resolution. For instance, the HS NMAD is reduced after correction by 50 % at a resolution of 180 m. The improvement is under 10 % at 20 m resolution as expected since the correction only dampers a low frequency signal. When the stable terrain residuals are used to compute Eq. (1) and (2) ($l_{cor}$ = 20 m, = 0.40 m), the modeled error is lower than the measured error. This is expected since the NMAD of the stable terrain residuals is lower than the NMAD of HS residual. However, the discrepancy between both models decreased at coarser resolution."*

Line 367-387: there is a justification being made here to bi-stereo imagery versus tri-stereo. Can you also articulate/quantify what the differences there may be in terms of cost differences (financial and computing). Will help to justify from a resource perspective why we should consider doing this all with pairs of images if it can articulated that there are significant savings terms of money and computing time/power requirements.
**A tri-stereo increases the cost by 50% since an additional image must be purchased (cost is proportional to imaged area). In addition larger areas can be imaged in bi-stereo configuration.**

**However, we have been told by Pléiades operator (Airbus D&S) that the tri-stereo mode does not significantly reduce the probability of a successful acquisition. Last, with our workflow, using tri-stereo images instead of bi-stereo images roughly double the computational time as the longest operation (disparity map calculation) occurs twice as many times.**

Line 410: can you clarify "decimetric accuracy"?
**We rephrased in L53:**
*"The results showed that snow depth could be retrieved from Pléiades images with an accuracy of roughly ~0.5 m (standard deviation of residuals 0.58 m for a pixel size of 2 m), suggesting that the method had the potential to become a viable alternative to airborne campaigns in mountain catchments with the benefits of a space based platform: access to any point on the globe and lower cost for the end-user."*

Line 421-424: merge with following paragraph?
**The structure of the paragraph in this part was largely modified.**

Line 430+: is this supposed to be a new paragraph?
**The structure of the paragraph in this part was largely modified.**

Line 436: "squares of length 210m". squares themselves don't have a length – can this be expressed differently?
**This sentence has been removed in the modification of the manuscript.**

Line 438: "unvalidated" -> "invalidated"?
**We modified according to the suggestion.**

Line 455: "probably the vast majority of mountain regions with seasonal snow cover"
-> "the vast majority of mountain regions with snow cover"
**We modified according to the suggestion.**

Line 458-459: "high competitions especially" -> "high tasking competition"
**We modified according to the suggestion.**

Line 459-260: can you identify specific satellite platforms that we should keep a look out for?
**Several very-high resolution stereo satellites are supposed to be launched in the years to come. Airbus, which operates Pléiades, advertises the launch of Pléiades-Neo in 2020 and 2022 (four satellites with image resolution of 30 cm). Maxar (former Digital Globe), which operates the WorldView fleet, should start the launch of the WorldView legion fleet in 2021 (at least six satellites with image resolution of 30 cm). The CNES, french space agency, is also working on the CO3D project in which four stereo satellites with 50 cm resolution should be launched in 2022.**
**There is no hint at which conditions images from these satellites would be available for research purpose.**
**We added a reference to the commercial project L502:**
*"More frequent acquisitions should, however, become easier as new stereo satellite fleets are to be launched in the coming years (Pléiades Neo, WorldView legion)."*

Line 471: this is an important point to make that needs more than 1 sentence at the end of the discussion. Can you emphasise the implications of this on where snow-depth mapping with this technique is valid and possible steps that may be available to address this limitation?

**We added a new paragraph in discussion to address specifically the topic of the applicability of snow-depth mapping with this technique in other regions (6.7. Generalization to other regions).**

Line 474-475: "satellite very high resolution stereo images" -> "high resolution stereo satellite images"
**We modified according to the suggestion.**

Figures: Information in the figures is good but the layout of the figure themselves need a fair bit of formatting work.

Figure 3, 5, 6, 7, 8, 9 , 10: Can you pull out the text/lines from inside the plotting areas in to legends (based on color) outside .
**We made the requested legend for figures 7, 9 and 10 but rather like to keep the other figures as it is to keep the figure more readable by color-blind readers.**

Figure 4: all of the lettering (referencing inset maps etc. ) is a little confusing in the legend. Perhaps add a upper-lower case distinction?
**We modified the labeling and the presentation of the lower panel to ease the reading of the figure.**

Figure 3, 7: Put x-axis descriptions outside of the plotting areas
**We modified according to the suggestion.**

Figure 7: slope residual plot quantiles exceed the plot area.
**As pointed, this leads some boxes not to be fully visible. These boxes represent very small portions of terrain (< 1 m²). We think that it is more beneficial to keep the axis adapted for the interpretation of the largest areas and to keep the same axis in all three subplots to ease comparison.**
**We now precise it in the caption of Fig. 7 (L610):**
*"Boxes where data were covering less than 1 km² are slightly transparent."*

Figure 8: could the a? panel have coordinates (UTM?) to provide scale and could then remove the scale bar. Also remove "Map" title and on b remove the titles and add x-axis label and units.
**We updated the figure according to the suggestion.**

Figure 9: "distance" -> "Distance"
**We updated the figure according to the suggestion.**

Figure 10: can you provide an explanation for "raw and "corrected" in the figure caption?
**We now avoid to use the term "raw" which is confusing but rather "before" and "after" correction. For instance in this caption L632:**
*"Empty blue circles are the NMAD of the residual HS maps averaged at different resolutions before the undulation correction. Filled red circles are the NMAD of the residual HS maps averaged at different resolutions after the undulation correction"*

References:
Bühler, Y., Adams, M. S., Bösch, R., Sto, A., Buhler, Y., Adams, M. S., Bosch, R. and Stoffel, A.: Mapping snow depth in alpine terrain with unmanned aerial systems (UASs): Potential and limitations, Cryosphere, 10, 1075–1088, doi:10.5194/tc-10-1075-2016, 2016.

Harder, P., Schirmer, M., Pomeroy, J. W. and Helgason, W. D.: Accuracy of snow depth estimation in mountain and prairie environments by an unmanned aerial vehicle, Cryosph., 10, 2559–2571, doi:10.5194/tc-10-2559-2016, 2016.

De Michele, C., Avanzi, F., Passoni, D., Barzaghi, R., Pinto, L., Dosso, P., Ghezzi, A., Gianatti, R. and Vedova, G. Della: Using a fixed-wing UAS to map snow depth distribution: An evaluation at peak accumulation, Cryosphere, 10(2), 511–522, doi:10.5194/tc-10-511-2016, 2016. Interactive comment on The Cryosphere Discuss., https://doi.org/10.5194/tc-2020-15, 2020.

---

## Author Comment (AC3) · 30 May 2020

**Answer to S. Fassnacht**

**The authors would like to thank Steven Fassnacht for his review. We provide below a point-by-point response to his comments and we explain how we intend to modify the manuscript in order to take them into account.**

This is an important paper as it presents the implementation of relatively new technology, stereo satellite imagery, to map snow. It provides the first comprehensive evaluation of a snow depth dataset derived from stereo satellite imagery by comparing it to an extensive lidar dataset (ASO). Overall the paper is presents important step towards better snow mapping.

The science is good and meaningful. The writing is not easy to read in many places.Specifically, the text could be more concise, and I recommend that the authors revisit how the paragraphs are structured and how sentences are written. While the English is not incorrect, the flow of the sentences makes it difficult to read. I acknowledge that the main authors are not native English speakers, but some of the 8 authors are native English speakers, and most of the authors have written extensively in English. I will use the Introduction as an example. It provides useful information, but is awkwardly written.The material reads partly like a stream of consciousness. There are new paragraphs that are a continuation of the previous paragraph. Please consider restructuring this section. For example, the text starting on lines 49-50 through line 73 presents the application of the stereoscopy using satellite imagery. Yet it starts in the middle of a paragraph with "The method was tested using two Pléiades stereo triplets over the Bassiès catchment in the Pyrenees (14.5 km²)." Then the authors tell us about what was done there. The next paragraph begins with "However" and is a continuation of the first paragraph. The last paragraph does present additional steps that will be seen in the rest of the paper. In restructuring the Introduction, end with objectives addressed or research questions posed so the reader knows where this paper is going.

There are various terms used that, while not incorrect, are awkward, such as on line 24 "deepen" in the context of a limited evaluation or on line 42 "decametric scale" total k about variability over 10s of meters. Similarly, some of the phrasing can be more succinct. For example, on lines 232-233 the authors state: "We evaluated the quality of the Pléiades HS maps over the area defined as the intersection of snow-covered terrain in Pléiades HS maps (snow mask) and ASO HS maps (HS greater than zero)."How about: "We evaluated the Pléiades HS maps for the area where both the Pléiades(snow mask) and ASO (HS greater than zero) HS maps had snow. This is stylistic, but some of the text is more complicated than it needs to be, and thus makes the paper more difficult to read.

**We worked on improving the structure of the manuscript, paragraphs and sentences. We modified the introduction so that it fits in four well structured paragraph. We moved some parts of the Method to keep its progression logic. We structured the Discussion in more homogeneous parts. The text of the revised manuscript was also carefully proofread by one coauthor who is a native english speaker to deal with sentence structure and terminology.**

Specific comments:

- Lines 77-78: "The snow-on Pléiades triplet was acquired 1st May 2017, the day before the ASO flight and close to the accumulation peak" How different is that snowpack overa day, i.e., how much does the snow depth vary between scene acquisitions?
**We could not access to local appropriate dataset (e.g. local weather station, consecutive ASO HS maps) within the time frame of this review to answer this question. From DeWalle and Rango (2008) we can expect a likely snow depth decrease of 2 cm d$^{-1}$ and a maximum of 8 cm d$^{-1}$.**
**Reference: D.R. DeWalle and A. Rango. 2008. Principles of snow hydrology. Cambridge, etc., Cambridge University Press. 410pp. ISBN 978-0-521-82362-3, hardback.**

- Line 84 and after: I assume that the word "stable terrain" means a location that is snow free? This should be explicitly stated.

**We tried to clarify this in the line above (L203):**

*"The co-registration vectors were calculated using the algorithm by Nuth and Kääb (2011) on areas where no elevation change is expected (i.e. stable terrain). The stable terrain areas, which are snow free terrain without trees, were determined by a supervised classification of the Pléiades multi-spectral images into a land cover map (see 4.1.5)."*

- Lines 26 versus 103 and 118: the abstract says "a snow-covered area of 137 km2"while the Study site section says "a 280 km2subzone." Which area is it? Line 141:what about the "excluded 25 km2" Line 306 states 138 km2

**The footprint of the images is 280 km² which is a subzone of the Tuolumne bassin covered by the ASO campaign. Parts of the subzone were not covered with snow (stable terrain, forest terrain) and parts were excluded (lakes, persistent snow in summer, 25 km² with artifacts in ASO maps). This leaves about 215 km² of snow-covered terrain of which remains 138 km² after snow mask buffering.**

**Thanks for noting the mistake. The snow covered area is now correctly stated as 138 km².**

- Lines 124-125 versus Table 1: The text states a resolution of 2 m, while Table 1 says 0.5 m, which is it?

**In the text we make the difference between the pan-chromatic images at a resolution of 0.5 m (L112) and the multi-spectral images at a resolution of 2 m (L116). We added this precision in the Table 1 (PAN: 0.5 m MS: 2.0 m) (L136).**

- The Methods are thorough. I suggest making sections 4.2 and 4.3 sub-sections of 4.1, as they are part of the overall Pléiades work-flow.- The authors evaluate snow covered (Pléiades snow mask HS maps versus ASO HS greater than zero HS maps), and stable terrain (snow free) areas. What about the omission and commission areas? At least provide the percent of the study area for each of these.

**Following the reviewer suggestion we modified the Method organisation. 4.2 and 4.3 (initial manuscript) are now 4.1.2 (L156) and 4.1.3 (L184) respectively. Some sentences were also moved to 4.1.1 to keep these three parts consistent.**

- Lines 240-260: it is unclear why these equations are presented here. The paragraph begins with "For hydrological applications, HS maps are often spatially aggregated, for example to calculate the amount of snow in a catchment or an elevation band." Either change this sentence or add a sentence so we know what these equations are used for.

**We inserted a sentence to clarify L243:**

*"The accuracy of HS maps is often discussed at (or close to) the highest resolution that is allowed by the sensor (e.g. Nolan et al. 2015, Marti et al., 2016). In practice however, HS maps may be subject to spatial averaging to assimilate in a snowpack model, to estimate catchment-scale HS or to compare with coarser satellite products and model output (Painter et al., 2016; Margulis et al., 2019; Shaw et al., 2019). The accuracy of the mean HS of a set of contiguous pixels is expected to be higher than a single pixel accuracy but depends on the spatial correlation of the errors (Rolstad et al., 2009). Hence, we performed an empirical assessment of the evolution of the accuracy of Pléiades HS as a function of resolution by aggregating the HS residual map to resolutions ranging between 3 m and 180 m (Berthier et al., 2016; Brun et al., 2017; Miles et al., 2018)."*

- Lines 263-265: it is not necessary to foreshadow what is in the Results "We first present the results for the HS maps calculated with the SGM-binary option and differentimage geometries. Then, we focus on the impact of the configuration of ASP. The best set of options and geometry is then used to analyze the spatial distribution of the residuals and to evaluate a model of the HS error." State clear objectives or research questions at the end of the Introduction and the reader would know what is to come.

**We removed these lines.**

- Tables 2 and 3: What is STD?

**We replaced STD with** *"standard deviation".*

- Figure 4: I'm not sure if you mean corniche or cornice?

**We replaced** *"corniche"* **with** *"cornice"* **(L595).**

- Line 287: it should be "Artifacts." Also, the phrase: "Artifacts of typically 20 m x 20 m..." is unclear

**We show examples of artifacts in the supplement and added a description L289:**

*"Patches of typically 20 m x 20 m with abnormally large HS (>10 m) compared to ASO (~3 m) are also observed with the Local-Search options around isolated trees. These artifacts are not visible with the SGM-binary or ternary options (Figure S1)."*

- Lines 306-307: what does "after erosion of the Pléiades snow mask" mean?

**Erosion is a term used in image morphology which can be indeed confusing. We replaced this sentence with (L221):**

*"The stable terrain and snow masks were shrunk (morphological erosion) with a radius of two pixels (4 m) and patches smaller than 30 pixels (270 m²) were removed."*

- Figure 9: it is unclear what the units in the y-axis are. The caption states: "h is the distance"

**We modified the axis labels and the caption of the figure:**

*"**Figure 9**. (a) Semi-variogram or spatial autocorrelation (□) against lag distance of the HS residuals (empty circles) and Pléiades elevation difference over stable terrain (filled circle). (b) Semi-variogram of the HS residuals for large distances before (blue line) and after correcting the undulation pattern (red line), illustrating the reduction in spatial variance at greater lag distances due to this correction (Sect. 6.5)."*

- This is also stylistic, but some sentence that tell what is upcoming and can be removed. For example, line 301-302 begins with "Figure 4 illustrates..." The authors can just tell us the key point(s) in the Figure as the caption tells the reader what the figure is.

**We simplified this sentence in (L302):**

*"The Pléiades HS map calculated with the selected image geometry and ASP configuration (front-nadir-back images, SGM-binary) compares well with the ASO HS map (Fig. 4)."*